# Critical exponents and scaling invariance in the absence of a critical point

N. Saratz[1], D.A. Zanin[1], U. Ramsperger[1], S.A. Cannas[2], D. Pescia[3] & A. Vindigni[1]

The paramagnetic-to-ferromagnetic phase transition is classified as a critical phenomenon due to the power-law behaviour shown by thermodynamic observables when the Curie point is approached. Here we report the observation of such a behaviour over extraordinarily many decades of suitable scaling variables in ultrathin Fe films, for certain ranges of temperature $T$ and applied field $B$. This despite the fact that the underlying critical point is practically unreachable because protected by a phase with a modulated domain structure, induced by the dipole–dipole interaction. The modulated structure has a well-defined spatial period and is realized in a portion of the $(T, B)$ plane that extends above the putative critical temperature, where thermodynamic quantities do not display any singularity. Our results imply that scaling behaviour of macroscopic observables is compatible with an avoided critical point.

[1] Laboratorium für Festkörperphysik, Eidgenössische Technische Hochschule Zürich, CH-8093 Zürich, Switzerland. [2] Facultad de Matemática, Astronomía y Física y Computacion, Universidad Nacional de Córdoba, Instituto de Física Enrique Gaviola (IFEG-CONICET), Ciudad Universitaria, 5000 Córdoba, Argentina. [3] Laboratorium für Festkörperphysik, Eidgenössische Technische Hochschule Zürich, and SIMDALEE2 Sources, Interaction with Matter, Detection and Analysis of Low Energy Electrons 2, Marie Sklodowska Curie FP7-PEOPLE-2013-ITN, CH-8093 Zürich, Switzerland. Correspondence and requests for materials should be addressed to A.V. (email: vindigni@phys.ethz.ch).

Magnetism, superconductivity and multiferroicity stand as examples of technologically relevant properties of matter resulting from a phase transition[1,2]. The conventional understanding of ferromagnetism is based on models of collective order in which atomic magnetic moments interact via one single leading—typically short-ranged—exchange interaction[3–6]. Yet, realistic ferromagnets suffer from the unavoidable presence of dipole–dipole interaction that, albeit weak, is long-ranged and frustrates the tendency to align magnetic moments parallel to each other promoted by the exchange interaction[7–10]. Probably, the most well-know consequence of this competition is the formation of magnetic domains, that is, a phase with modulated magnetization at low temperature. In the early 1980s, Wasilevsky studied the loss of modulated magnetic order with increasing temperature for an idealized sample geometry and within the mean-field (MF) approximation. He showed that the fluctuations of the local magnetization above the MF Curie temperature $T_c^{MF}$ are also modulated and their spatial period matches exactly the width of domains entering the sample below $T_c^{MF}$. With his own words: '… we can conclude that even in the paramagnetic phase [above $T_c^{MF}$]… the demagnetizing field induces a domainization effect of the fluctuations of magnetization'[11,12]. It is important to point out that this domainization effect produced by the dipolar field is not to be confused with the droplets that define the spatial extent of statistical fluctuations by means of the correlation length in the vicinity of an ordinary critical point[3]. In the prevailing understanding of ferromagnetism—and of second-order phase transitions in general—based on renormalization group, the divergence of the correlation length lies behind scaling behaviour and universality of critical exponents[3–6]. Within the theoretical scenario depicted by Wasilevsky the critical point of a ferromagnet is replaced by a low-temperature phase with modulated magnetization that persists above $T_c^{MF}$. In particular, the spatial period of the modulated phase provides, alongside the correlation length, a second long length scale that transforms the critical point into a so-called avoided critical point. The more general theoretical framework of avoided criticality was developed in the 1990s (refs 13–15) and foresees that even a small amount of frustration changes the critical point of an unfrustrated model of collective order into a completely different object—generally not a second-order transition point—depending on details like the spatial range and the strength of the competing interactions involved, the dimensionality of the system and the number of components of the order parameter[16–22]. A non-exhaustive list of pattern-forming systems for which the phenomenology of avoided criticality has been proposed comprises magnetic films, ferrofluids, diblock copolymers, amphiphilic solutions, systems undergoing Turing-like phase separating chemical reactions and charged stripes in cuprate high-$T_c$ superconductors[22–26].

Here we report two novel aspects of the ferromagnetic phase transition, observed in perpendicularly magnetized ultrathin films of Fe grown on Cu(001). First, partially in line Wasilevsky's hypothesis[11,12], we found that the width of stripe domains observed at low temperature remains finite at a presumed $T_c$—not simply defined as the MF critical temperature introduced by Wasilevsky—and evolves continuously with temperature, up to at least 30° beyond $T_c$ (that is, 10% of $T_c$ itself), without displaying any singularity. (Note that ref. 22 predicts a divergence of the modulation length at some temperature.) This appears to represent a kind of conservation law for the spatial period of modulation in pattern-forming systems. Furthermore, it implies that the divergence of the susceptibility as a function of temperature expected for a standard ferromagnet at $T_c$ is replaced by an analytic behaviour. Second, we observe that the conventional critical point itself is indeed eliminated by the

presence of domains in line with the avoided-criticality scenario. However, the most distinctive feature of a second-order phase transition, namely the conventional scaling à la Kadanoff[3–6], is recovered in a temperature and magnetic-field range sufficiently away from the presumed critical point, over up to 80 orders of magnitude of the suitable scaling variable. In this respect—but only in this respect—the system behaves as if the critical point existed.

## Results

**The model Hamiltonian.** Since the most relevant experimental outcomes of this work result from the analysis of the global magnetization as a function of the applied magnetic field $B$ and temperature $T$, it is convenient to define from the outset the theoretical framework underlying this analysis. We consider the Hamiltonian[27,28]

$$\mathscr{H} = -J \sum_{\langle i,j \rangle} S_i S_j + g \sum_{(i,j)} \frac{S_i S_j}{r_{ij}^\alpha} - b \sum_i S_i \qquad (1)$$

where $S_i = \pm 1$ are Ising spin variables disposed on a two-dimensional (2D) lattice and associated with the two out-of-plane directions along which magnetic moments preferentially point. The first term on the right-hand side of equation (1) represents the exchange interaction (the sum runs over all pairs of nearest-neighbouring sites), while the second term represents a long-ranged frustrating interaction decaying with a power $\alpha$ of the distance between spin pairs. The third term represents the Zeeman energy with $b = \mu \cdot B$, $\mu$ being the atomic magnetic moment. The equilibrium magnetization is defined as $m \doteq \sum \langle S_i \rangle / N$, $N$ being the total number of spins and $\langle S_i \rangle$ the statistical average.

Let us first recall the essential aspects of the unfrustrated model ($g = 0$). The familiar description of the ferromagnetic phase transition foresees that $m(T, b)$ develops a discontinuity at $b = 0$ below the Curie temperature, which defines a situation of spontaneously broken symmetry[1] (black curves in Fig. 1a). Above the Curie temperature $m(T, b)$ shows no discontinuity and behaves linearly for small enough fields (red curve in Fig. 1a). This behaviour is typically predicted by models in which ferromagnetic order is stabilized by a short-ranged exchange interaction, favouring parallel alignment of neighbouring spins[3–6]. In the absence of field ($b = 0$), configurations obtained by flipping all the spins have the same energy, that is, the Hamiltonian possesses a $\mathbb{Z}_2$ symmetry (the symmetry elements being the identity and the simultaneous change of sign of all variables). The minimal energy is realized by the two configurations with all the spin pointing along the same direction (Fig. 1b). Passing from one of such configurations to the other requires the creation of a domain wall. For finite temperature and finite fields, the discontinuity obtained for $T < T_c$ specifically reflects the spontaneous breaking of the $\mathbb{Z}_2$ symmetry[3,29]. In the neighbourhood of the transition point $(T, b) = (T_c, 0)$ observables behave critically. One implication of criticality is that the equation of state $m$ as a function of $(T, b)$ has a scaling representation[30,31]:

$$\frac{m}{|\tau|^\beta} = g_\pm \left( \frac{b}{|\tau|^{\beta\delta}} \right) \qquad (2)$$

with $g_\pm(x)$ being scaling functions, $\tau = (T - T_c)/T_c$ the reduced temperature ($g_+$ and $g_-$ referring to positive and negative values, respectively, of $\tau$)[31]. $\beta$ and $\delta$ are two critical exponents characteristic of a specific universality class, the 2D-Ising universality class for the model defined by Hamiltonian (1) with $g = 0$. Another implication of criticality is that $m(T, b)$ and the magnetic susceptibility $\chi(T, b) \doteq \frac{\partial m}{\partial b} \big|_{b=0}$—as well as other

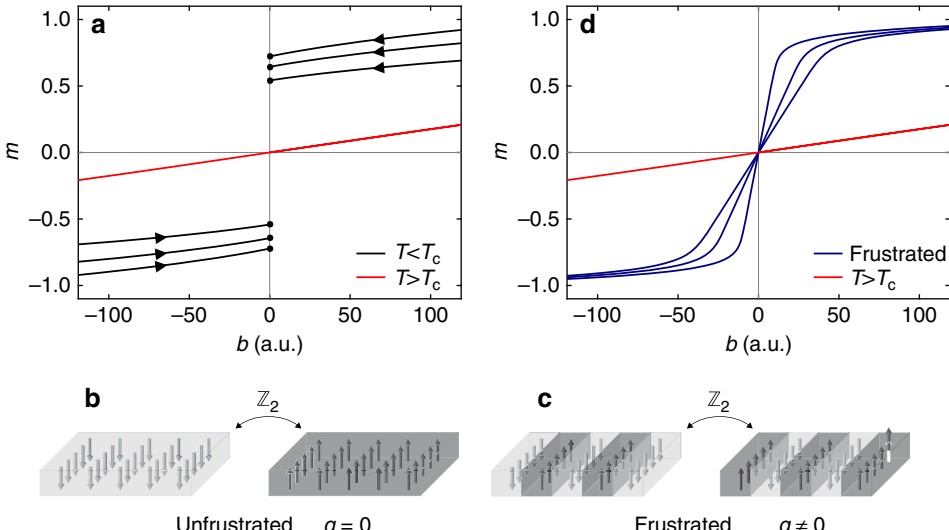

**Figure 1 | Properties of the model Hamiltonian.** (**a**) Sketch of some equilibrium magnetization curves $m(T, b)$ as a function of the applied field $b$ theoretically expected for an unfrustrated ferromagnet below $T_c$ (black lines). The equilibrium curves are described coming from large fields, as indicated by arrows. Note the jump when $b = 0$ is crossed for $T < T_c$, highlighted by dots. A pictorial curve representing the paramagnetic behaviour, $T > T_c$, is also displayed (red line). (**b**) Two ground-state configurations (linked by the $\mathbb{Z}_2$ symmetry) of the model defined by Hamiltonian (1) in the absence ($g = 0$) and (**c**) in the presence ($g \neq 0$) of frustration. (**d**) Sketch of some equilibrium magnetization curves $m$ as a function of the applied field $b$ for a frustrated system at different temperatures (blue lines). The discontinuity expected in the unfrustrated case at $b = 0$ gives way to a linear behaviour, as a result of the frustrating effect of dipolar interaction. The red line is the same as in the plot **a**.

observables not considered in this study—become power-law functions of $\tau$ and $b$ while approaching the critical point.

In the presence of frustration ($g \neq 0$), for $\alpha \leq 3$ the configuration with lowest energy is not the ferromagnetic phase but rather a modulated phase, typically striped[32,33] (Fig. 1c). In this case, configurations obtained by a rigid translation of the whole striped pattern have the same energy. Concrete realizations of this model involve either Coulomb repulsion ($\alpha = 1$) or dipole–dipole antiferromagnetic interaction ($\alpha = 3$) that competes with a ferromagnetic short-ranged interaction[22]. This competition leads to the formation of a striped ground state whose elementary excitations are described by an elastic-like Hamiltonian associated with the displacement of domain walls—as a result of the subtle interplay between the two interactions[17,21]. The experimental results presented in the next subsections refer to a ferromagnetic model system representative of the Hamiltonian (1) with $\alpha = 3$ (refs 18,19,27,33–36). Some unconventional behaviours observed in these ferromagnetic films, such as the systematic occurrence of inverse-symmetry-breaking transitions of magnetic-domain patterns, have been explained theoretically starting from the Hamiltonian (1)[28,37]. In the limit of $J \gg g$, relevant for the experimental system, the spectrum of the elastic-like excitations mentioned above is gapless. In particular, a zero-energy mode (Goldstone mode) connects any pairs of stripe patterns related by a $\mathbb{Z}_2$ symmetry operation (Fig. 1c). Therefore, from the perspective of elementary excitations, the frustrated model is more akin to systems with continuous symmetry rather than to the unfrustrated Ising model with discrete $\mathbb{Z}_2$ symmetry—whose elementary excitations are domain walls with a finite energy. These differences suggest that the loss of the magnetic order realized in the ground state proceeds at finite temperature in a completely different way with respect to the unfrustrated case: a phase with finite spontaneous magnetization ($m(T, b = 0) \neq 0$) is never realized at thermodynamic equilibrium and the system passes from a phase with domain patterns and global vanishing magnetization to the paramagnetic phase without showing any singularity related to the breaking of the discrete $\mathbb{Z}_2$ symmetry. As

a function of $b$, $m$ does not develop any discontinuity, neither below nor above the transition temperature (Fig. 1d). The label 'avoided criticality' is used to underline these major differences with respect to the unfrustrated model.

The low-temperature patterned phase has been investigated theoretically at different levels of approximation and focusing on aspects that range from self-generated glassiness[14,15,20,22] to topological phase transitions[17–19,27,33–36]. All the literature we are aware of supports the absence of long-range positional order of domains at finite temperature[17,18,21], thus suggesting that not even a kind of staggered magnetization associated with the striped pattern is expected to display the 2D-Ising critical behaviour (nor the MF critical behaviour foreseen by Wasilevsky[11,12]). In the prevailing understanding of critical phenomena the fulfilment of equation (2) and other scaling relations is associated directly with spontaneous breaking of a specific symmetry of the Hamiltonian in $b = 0$. Since the arguments given above exclude spontaneous breaking of the $\mathbb{Z}_2$ symmetry in the frustrated model, one should not expect to observe scaling relations with Ising critical exponents when $g \neq 0$. With the present study we, instead, demonstrate that the critical scaling laws obeyed by the unfrustrated model are actually recovered in the frustrated case sufficiently away from a putative critical point: numerical and experimental evidence supporting this statement is provided in the following.

**Monte Carlo simulations.** We performed a series of Monte Carlo simulations using the Hamiltonian (1) with $\alpha = 3$ (see the subsection 'Technical details of Monte Carlo simulations' in Methods). Several theoretical works investigated the modulated phases of this model[27,33–36]. In this study we explored the transition between those modulated phases and the phase with uniform magnetization, specifically searching for a region of the $(T, b)$ plane, where the equations of state of ferromagnets were obeyed. The resulting magnetization curves are plotted in Fig. 2a and their scaling representation (2) in Fig. 2b. In Fig. 2b the upper

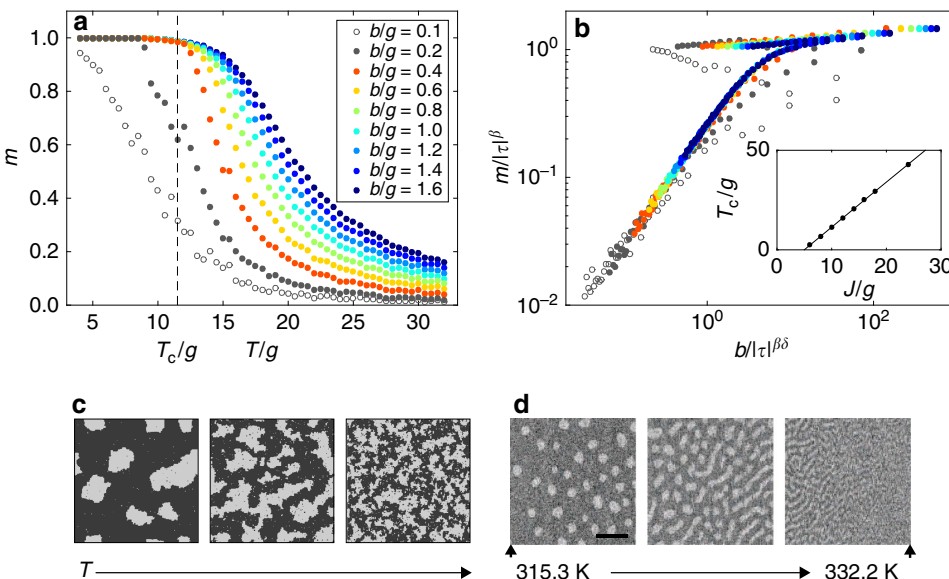

**Figure 2 | Monte Carlo simulations. (a)** A family of $m(T)$ isochamps obtained from Monte Carlo simulations for $J/g = 10$ ($k_B = 1$ is assumed), $L_x = L_y = 120$ and different values of the Zeeman energy $b = \mu \cdot B$. **(b)** Scaling plot $m/|\tau|^\beta$ versus $b/|\tau|^{\beta\delta}$ for the data shown in **a** using the 2D-Ising critical exponents: collapsing is realized only when for $b > b_c$ (see the subsection 'Technical details of Monte Carlo simulations' in Methods). Inset: the characteristic temperature $T_c$, used to optimize data collapsing in the Monte Carlo scaling plots, as a function of $J/g$. **(c)** Representative snapshots of Monte Carlo configurations obtained for $L_x = L_y = 200$, $J/g = 10$, $b/g = 0.1$ and $T/g = 10, 13, 16$ (from left to right). **(d)** Sequence of selected SEMPA images recorded on an Fe film of 1.9 MLs deposited on Cu(001) while increasing the temperature from 315.3 to 332.2 K in a constant field $B = 0.61 \times 10^{-4}$ T. The length of the bar is 10 μm.

branch contains the Monte Carlo data for negative values of $\tau$, the lower branch those for positive values of $\tau$ (the two branches meet at large values of the variable on the horizontal axis). Note that the resulting $T_c$ as a function of $J/g$ can be expressed as the transition temperature of the 2D-Ising model without frustration, $T_c^{\text{Ising}} = 2J/\ln(1 + \sqrt{2})$ (assuming $k_B = 1$), subtracted by a value which corresponds to about $11.2g$ (inset of Fig. 2b). This suggests that in the limit of vanishing frustration ($g \to 0$), the temperature $T_c$ tends to $T_c^{\text{Ising}}$. Collapsing could be indeed realized for $b > b_c$ (coloured points in Fig. 2b), $b_c$ being a threshold field above which the system is in a uniformly magnetized phase (see the subsection 'Technical details of Monte Carlo simulations' in Methods). Significant departure from collapsing was, instead, observed for $b < b_c$ (open and full grey points corresponding to $b/g = 0.1$ and 0.2), that is, inside the region of the $(T, b)$ plane where a phase with modulated magnetization appears. Snapshots of representative Monte Carlo configurations computed for $b/g = 0.1$ are given in Fig. 2c, along with selected magnetic configurations observed in the experimental system (Fig. 2d). Note that the magnetization curves obtained for $b > b_c$ display qualitatively the same spreading with increasing fields expected for the unfrustrated model. The picture emerging from Fig. 2a,b underlines that scaling and power laws of the Ising type are also compatible with a scenario in which the $\mathbb{Z}_2$ symmetry is not spontaneously broken in the magnetization of any sublattice. Henceforth, we will use the wording 'ferromagnetic scaling region' to indicate the region of the $(T, b)$ parameter space in which equations of state of standard ferromagnets are obeyed even if a Curie transition point ($T_c$, $b = 0$), in the conventional understanding, is not realized.

We now summarize the scenario emerging from the Monte Carlo simulations. For large enough magnetic fields, the magnetization as a function of $T$ and $B$ should obey a scaling relation such as the one given in equation (2). This is the ferromagnetic scaling region, delimited schematically by a dashed violet line in Fig. 3a. The shape of the dashed line is not

compelling from the Monte Carlo simulations. For convenience, we draw it in line with the experimental results presented in subsection 'Experimental phase diagram'. For small enough fields magnetic films are expected to be in the modulated phase, in which ferromagnetic scaling should not be obeyed anymore (dark grey in Fig. 3a). Instead, the relation between the magnetization and the applied field $B$ should be a linear one, that is, the graph of the experimental magnetization as a function of magnetic field at finite temperatures should be qualitatively similar to the ones sketched in Fig. 1d, rather than the ones obtained in the absence of frustration (Fig. 1a). Further information about the expected phase diagram are sketched in Fig. 3a on the basis of previous experimental studies (see the subsection on 'Experimental scaling plots').

**Experimental scaling plots.** The ultrathin Fe films investigated in this work are grown at room temperature by molecular-beam epitaxy onto the (001) surface of a Cu single crystal (see refs 38–40 for details). The samples extend macroscopically along the directions defining the film plane and have a thickness between 1.6 and 2.0 atomic monolayers (MLs). For this range of thickness these films possess an out-of-plane magnetocrystalline anisotropy—strong enough to overcome the shape anisotropy—so that magnetic moments preferentially point perpendicularly to the plane[41]. In this sense, these films represent an experimental counterpart of Hamiltonian (1) with $\alpha = 3$. Typically, modulated phases in similar experimental systems consist of stripes and/or bubble domains of opposite magnetization[42–44] (some representative images are shown in Fig. 2d and later (Fig. 6a–f)). In previous studies magnetic imaging of these films was performed with scanning electron microscopy with polarization analysis (SEMPA)[39,40,45], see the subsection 'SEMPA imaging' in Methods and Fig. 3b. SEMPA revealed, on one side, that above a temperature $T^*$ (sketched in Fig. 3a) domains become mobile[44–47]. On the other side, below $T^*$ static domains were

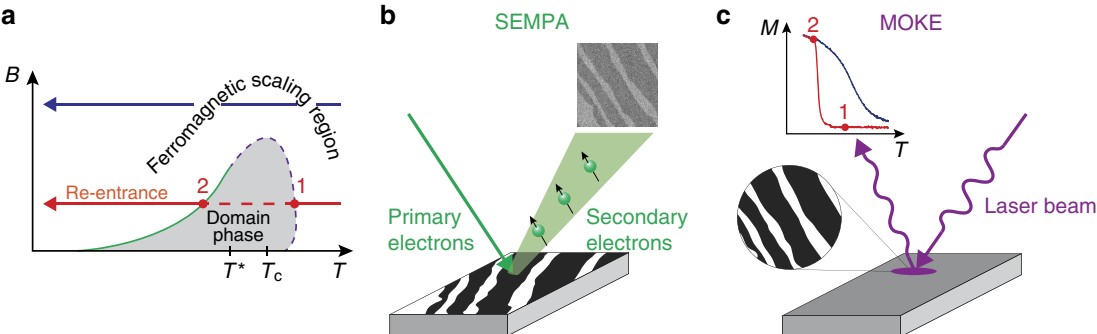

**Figure 3 | Illustration of experimental methods.** (**a**) Phase diagram of a frustrated perpendicularly magnetized film in the $(T, B)$ parameter space. Dashed violet line and green line, the points 1 and 2 and the horizontal red and blue lines are explained in the main text. The portion of the $(T, B)$ plane in which the scaling equations of state of ferromagnets are obeyed is indicated as ferromagnetic scaling region. (**b**) In SEMPA, the spatially focussed primary electron beam is scanned across the sample carrying magnetic domains (dark and bright stripes). The secondary electrons ejected from the surface (green) are spin polarized and are used for imaging the domains. (**c**) The MOKE experiment averages over all domains within the focus of the light beam (the violet circle, about 1 mm$^2$). Measuring with MOKE the magnetization as a function of $T$ along the blue and red lines (**a**) produces the $M(T)$ graphs in the inset given with the same colours.

observed that transformed into a uniform state on cooling (point '2' along the re-entrance red horizontal line in Fig. 3a). To complement the information about magnetic phases deduced from SEMPA imaging, the present work adds the detection of the spatially averaged magnetization as a function of $B$ and $T$ (Fig. 3c), measured using the magneto-optical Kerr effect (MOKE)[48] (A full magnetization curve at constant $T$ can be recorded with MOKE within seconds or less, while the time required to scan a SEMPA image is in the range of minutes.) We anticipate that analysing the MOKE data we were able to determine experimentally the line sketched as dashed violet in Fig. 3a and to locate the putative critical point at $(T, B) = (T_c, 0)$ in spite of the fact that it fell in the region occupied by domains (Fig. 3a). Technically speaking, MOKE (Fig. 3c) measures a signal that is proportional to the spatially averaged magnetization within the size of the light beam (typically about 1 mm$^2$)[48] and is given in arbitrary units: we denote the measured quantity with the letter $M$. From now on, all quantities involving $M$ must be considered in arbitrary units. On the basis of our Monte Carlo simulations, we expect two types of $M(T)$ curves, depending on whether we vary the temperature in a situation of strong (dark blue horizontal line in Fig. 3a) or weak (red horizontal line in Fig. 3a) magnetic field. Along the blue line, the film is in a state of uniform magnetization and the $M(T)$ curve follows the blue graph in the inset of Fig. 3c. Along the red line, at low temperatures, on the left-hand side of the green line in Fig. 3a a small magnetic field is enough to establish the uniform magnetic state and $M$ easily saturates. When the temperature is increased and the green line in the $(T, B)$ parameter space is crossed (point 2 in Fig. 3a), the macroscopic magnetization abruptly drops to almost zero as the system enters the phase of static modulated order: the local magnetization within the domains is still large but $M$ almost vanishes because of the cancellation of finite opposite values of the local magnetization within the domains. The corresponding $M(T)$ is sketched in the inset of Fig. 3c with the same colour. At the point 1 (Fig. 3a,c) the $M(T)$ is small and varies smoothly.

Figure 4a shows a measured $M(T)$ family of isochamps for a film with thickness of 1.75 MLs. The magnetic field was swept with a frequency varying between $10^{-1}$ and 1 Hz. $M(B)$ curves measured at fixed temperature within this range of frequencies coincide, therefore we assume that we are observing properties related to thermodynamic equilibrium. At higher frequencies the shape of the magnetization curves depended of the sweeping rate

of $B$. Results of these studies will be reported in a separate paper. We distinguish two extreme sets of $M(T)$ curves. On the right-hand side those corresponding to larger values of $B$, an example of which is highlighted in the data with a dashed orange line (the colour code used to label the values of $B$ is indicated along the vertical bar in Fig. 4b). The horizontal line in the $(T, B)$ parameter space, along which the orange $M(T)$ curve was taken, is given in orange in the inset of Fig. 4a. It resides above the blue dotted curve, determined as explained in the subsection 'Experimental determination of $T_c$' in Methods, and belongs to the set of curves illustrated in Fig. 3a by the horizontal blue line. This means that for these values of $(T, B)$ the film is in the uniformly magnetized phase and, accordingly, the familiar behaviour of $M(T)$ curves separating out as $B$ is increased, that is, moving away from the critical point, appears. On the left-hand side of Fig. 4a one finds the family of curves corresponding to low $B$, an example of which is highlighted in the data with a thick, red line also shown in the inset of Fig. 3a (and illustrated by the red graph in the inset to Fig. 3c). At low temperatures, on the left-hand side of the blue dotted line, a small magnetic field is enough to establish the uniform magnetic phase and $M$ easily saturates. When the temperature is increased and the blue line in the $(T, B)$ parameter space is crossed, the macroscopic magnetization abruptly drops to almost zero as the system enters the phase of static modulated order: the local magnetization within the domains is still large but $M$ almost vanishes because of the cancellation of finite opposite values of the local magnetization within the domains (this transition is marked with '2' in the sketch of Fig. 3a).

To produce the scaling plots of the data in Fig. 4a we determined $T_c = 300 \pm 1$ K (not to be confused with $T^*$), and the critical exponents $\beta = 0.15 \pm 0.03$, $\delta = 13 \pm 2$ usually defined in the vicinity of an ordinary critical point[3–6] (see the subsections 'Experimental determination of $T_c$' and 'Experimental determination of $\beta$ and $\delta$' in Methods). The values of the critical exponents $\beta$ and $\delta$ found experimentally are close to those expected for the unfrustrated 2D-Ising universality class[49–51]. (We point out that some models of three-dimensional ferromagnetism[49–51] predict an ordinary critical point in the presence of the dipolar interaction, with critical exponents computable, for example, within the Renormalization Group approach.) Notice that in the simulations the 2D-Ising critical exponents were assumed. Here we obtain them from the experimental data. In Fig. 4b a scaling plot is attempted with

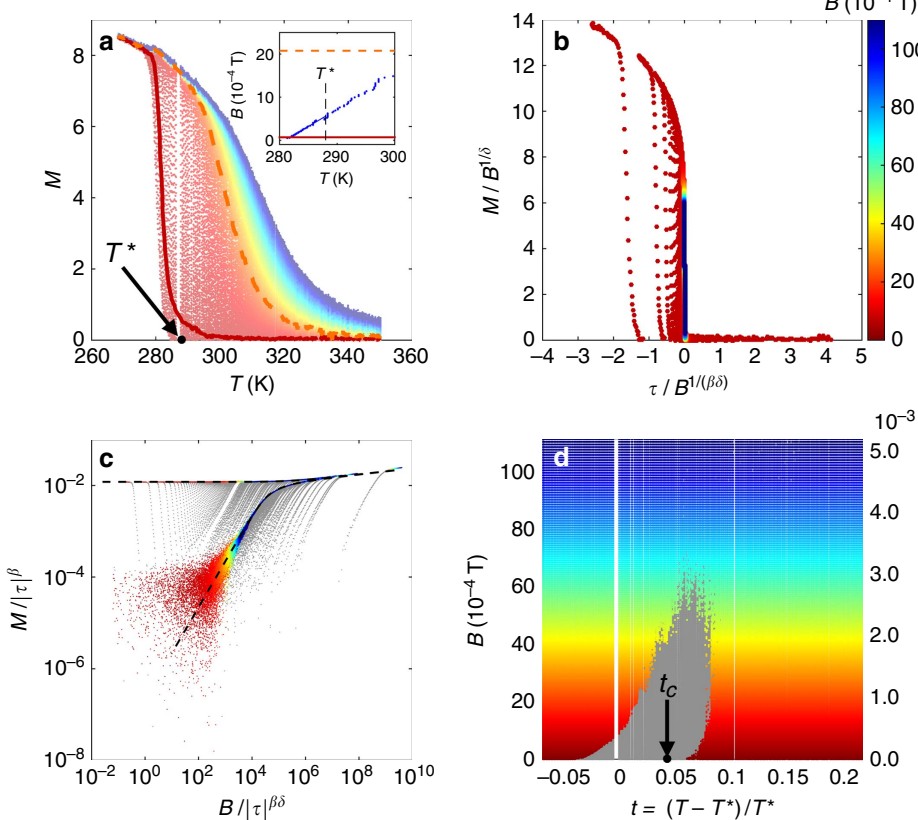

**Figure 4 | Scaling plots and experimental phase diagram.** (**a**) Family of experimental $M(T)$ isochamps (about $10^5$ experimental data points). The film thickness is about 1.75 MLs. The temperature $T^*$ is defined in the main text. Inset: paths on the $(T,B)$ plane corresponding to the two $M(T)$ curves highlighted in the main frame. Blue dots mark the boundary between the modulated and the uniform phase. (**b**) Scaling plot $M/B^{1/\delta}$ versus $\tau/B^{1/(\beta\delta)}$. The colour code used for the applied magnetic field $B$ in all the plots is given along the vertical bar. (**c**) Scaling plot $M/|\tau|^\beta$ versus $B/|\tau|^{\beta\delta}$. The dashed line represents the scaling function of the 2D-Ising model taken from ref. 31. (**d**) The same experimental data points are transferred within the $(T, B)$ parameter space. $t_c$ corresponds to the putative critical temperature. The vertical scale on the right-hand side gives the values of the magnetic field $B$ in units of the saturation magnetization of Fe, 2.16 T. The thick white vertical line and few thin white horizontal lines are due to the failure of data recording during data taking.

the variables $M/B^{1/\delta}$ (vertical axis) and $\tau/B^{1/(\beta\delta)}$ (horizontal axis)[52]. Except for a (substantial) set of data points, we observe that the spread of curves inherent to Fig. 4a has reduced considerably, that is, many $M(T)$ isochamps have collapsed onto one thin graph in Fig. 4b. As anticipated, such a collapsing is considered to be a signature of scaling invariance realized while approaching the critical point $(T, B) = (T_c, 0)$. But the emphasis must be on the set of non-collapsed data points: they are better evidenced in the scaling representation given in equation (2)—replacing $m$ with $M$ and $b$ with $B$—plotted in Fig. 4c. The collapsed and non-collapsed data points were retrieved using an *ad hoc* software that distinguished the high-density points—which we consider as collapsed—from the low-density ones—which we consider as non-collapsed. The collapsed data points, which build the majority, are rendered with their original colours and are observed to be distributed onto the graphs of the 2D-Ising scaling functions $g_-(x)$ (dashed line, upper branch) and $g_+(x)$ (dashed line, lower branch)[31]. The non-collapsed data points are rendered in grey and are observed to fall outside the graphs of the 2D-Ising scaling functions.

**Experimental phase diagram**. The coloured (collapsed) and grey (non-collapsed data points, violating scaling) were subsequently transferred into their place in the $(T, B)$ plane in Fig. 4d, where

they appear outside and, respectively, inside a bell-shaped region. As anticipated, this region, called the grey zone henceforth, surrounds and protects the putative critical point $(T_c, 0)$. The bell-shaped region starts on the left with the boundary line marking the transition from static modulated phase to the uniform phase (the blue line in the inset of Fig. 4a up to $t = 0$). The phase diagram obtained by SEMPA in a previous publication[39] corresponds to this portion of the grey zone, residing below $t = 0$. Slightly above $t = 0$ stripes are observed to become mobile[45]. The grey zone continues well beyond $t = 0$: it is conceivable that within the entire grey zone, where scaling is violated, some kind of modulated phase exists.

**Further representations of scaling**. In Fig. 5a the same data points shown in Fig. 4a are plotted in the Griffiths–Widom representation

$$\frac{B}{M^\delta} = f\left(\tau/M^{1/\beta}\right) \qquad (3)$$

with $f(x)$ being a suitable scaling function[30,31] with the same colour coding as in Fig. 4c. Particularly clear is how grey points deviate from scaling for the data plotted in the inset $(-2 \times 10^{16} \le x \le 10^{17}$, with $x = \tau/M^{1/\beta})$. In the main frame the non-collapsed, grey points decorate the coloured line of clearly collapsed points up to $x \sim 10^{23}$. For larger values of $x$, the line of

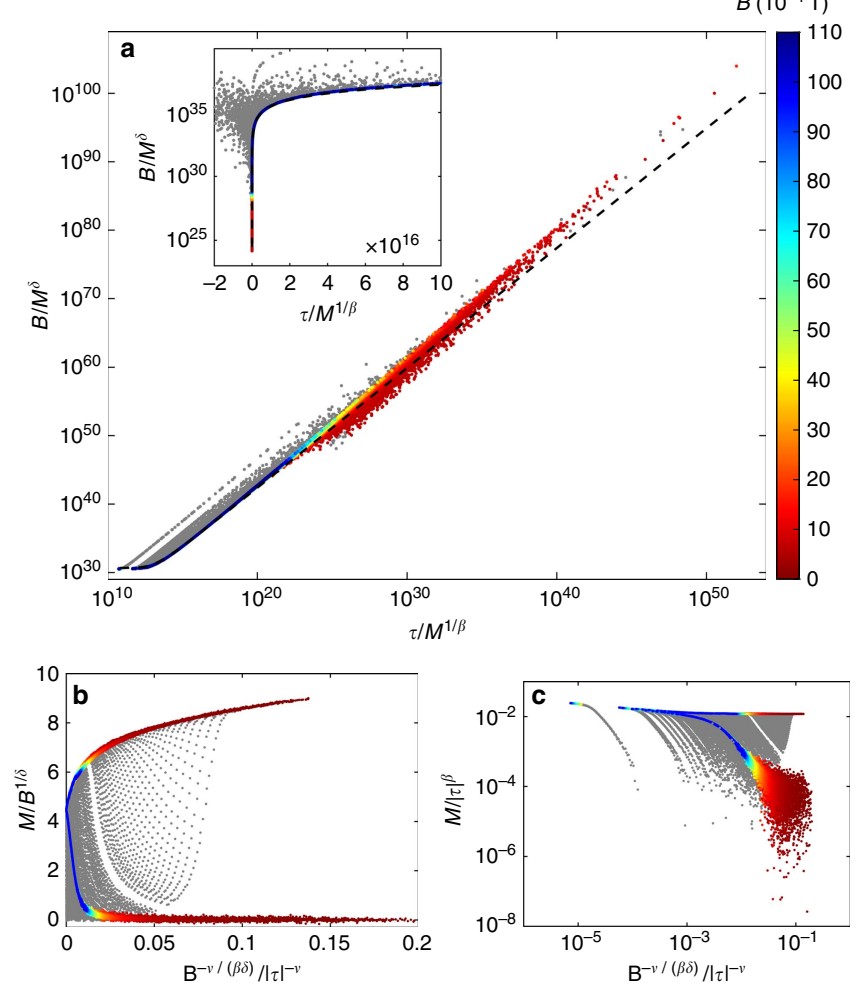

**Figure 5 | Further representations of scaling. (a)** The Griffiths–Widom representation of equation (3) uses the scaling variables $B/M^\delta$ versus $x = \tau/M^{1/\beta}$. The scale is log–log in the main panel and log–linear in the inset. The dashed line represents the theoretical scaling function given in ref. 31 for the 2D-Ising model (shifted by non-universal, constant scaling factors). Note that collapsed data points overlap to this line throughout the range $-6 \times 10^{12} \lessgtr x \lessgtr 10^{23}$. The slight deviations for $x > 10^{40}$ are due to the experimental value of the exponent $\gamma$ being slightly different from the one expected for the 2D-Ising model (remember that $\gamma$ is the slope of the graph for very large values of $x$). **(b)** Representation of scaling given in equation (7) using the same data points shown in Fig. 4c, with the same colour coding. Coloured points are along the scaling function $F_1(x)$. **(c)** Representation of scaling given in equation (8) using the same data points shown in Fig. 4c, with the same colour coding. Coloured points are along the scaling function $F_2(x)$.

coloured dots broadens and deviates from the theoretical scaling function of the unfrustrated 2D-Ising model (dashed line) as well. In this Griffiths–Widom representation data collapsing is realized over 40 orders of magnitude with respect to the $x$-variable defined above and 80 orders of magnitude with respect to the variable $B/M^\delta$ (note the remarkable agreement in the inset). Further representations of scaling are found in Fig. 5b,c (see the subsection 'Scaling and characteristic lengths' in Methods for explanation).

**Experimental stripe width**. A typical evolution of magnetic-domain patterns as a function of temperature in $B = 0$ imaged with SEMPA is shown in Fig. 6a–f. The spatial resolution of the present SEMPA instrument is in the submicrometre range. Dark- and light-grey regions correspond to domains with opposite perpendicular magnetization. The in-plane components of the magnetization vanish within the experimental sensitivity. The images refer to a Fe film with thickness of 2 MLs. The temperature is lowered while the image is acquired, starting from a to f. The limiting temperatures are always indicated at the edges

of each image and temperatures in-between are interpolated linearly. The left-hand side of the image in Fig. 6a appears almost contrastless, while on the right of the red line—which defines the temperature $T^\star = 333 \pm 1\,K$—a very weak, stripe-like contrast develops. Further right, approaching the edge of the same image, the contrast becomes strong enough to allow the determination of the stripe width $L(T)$. The transition from static to mobile stripes occurring at $T^\star$ (red line in Fig. 6a) was already investigated in previous works[44–47]. It is not clear yet how to call the phase with mobile stripes. Possibly the most appropriate definition is that of stripe liquid[20] or floating solid[53], which highlight the lack of positional order in this phase[21]. With certainty, the temperature corresponding to the red line cannot be identified with a Curie temperature. This in spite of the fact that when $T^\star$ is approached from below the local magnetization inside the stripes decreases sizeably, mimicking the decrease of the spontaneous magnetization in a uniformly magnetized phase close to $T_c$ (refs 39,45) (see the subsection 'Magnetization within domains' in Methods).

The magenta squares in Fig. 6g correspond to the values of $L$ determined by direct inspection of SEMPA images at the

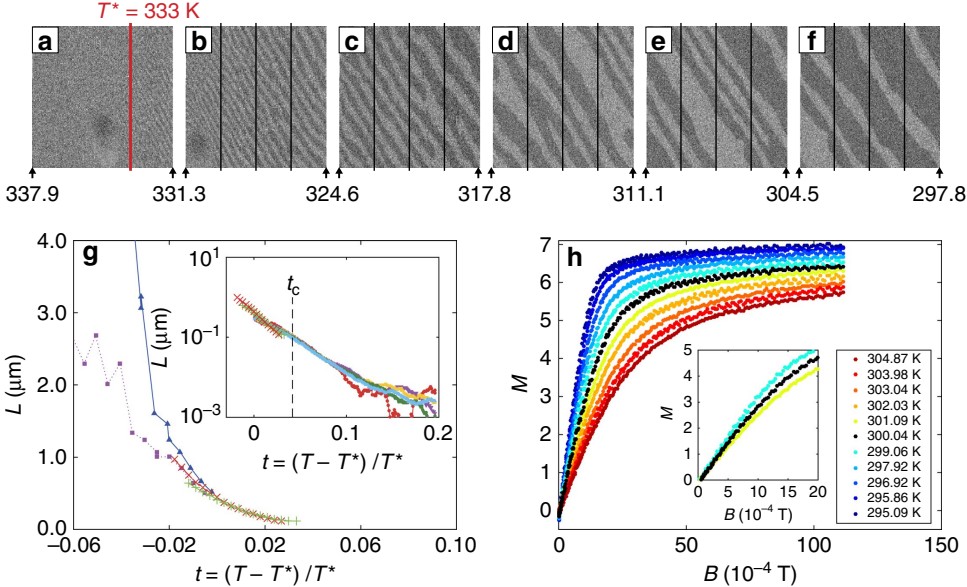

**Figure 6 | Stripe width as a function of temperature.** (**a–f**) SEMPA images of the spatially resolved magnetization taken while cooling: temperature is decreased from **a** to **f**. The vertical lines indicate the temperatures in K at which the values of the stripe width were extracted from each image and reported in the (**g**) plot. Images are taken in nominally zero applied field and their linear size is 22 μm. (**g**) Stripe width $L$ as a function of the reduced temperature $t$ for a film with thickness of 2 MLs ($d = 3.6 \times 10^{-4}$ μm) obtained by visual inspection of SEMPA images (blue triangles and magenta squares) and from equation (4) using the total magnetization measured with MOKE (green and red crosses). Inset: data produced applying equation (4) on the $M(T,B)$ data for $B = 0.5, 1.3, 2.5, 4.1$ and $6.0 \times 10^{-4}$ T (film thickness 1.75 MLs, $d = 3.15 \times 10^{-4}$ μm) are plotted in a log–linear scale together with the green and red data points of the main frame, measured on a different film ($d = 3.6 \times 10^{-4}$ μm). Owing to the linear relation between $M$ and $B$ within the grey zone, the graphs for different values of the field practically coincide. (**h**) $M$ versus $B$ curves for temperatures close to $T_c$. The inset shows a zoom of the small-field region for three selected temperatures, the black one corresponding to the putative critical isotherm. The small offset, of $< 0.5 10^{-4}$ T, visible in the inset is due to a small residual magnetic field for the nominal value of $B = 0$.

temperatures ($T < T^*$) indicated by vertical lines in Fig. 6a–f. The horizontal scale gives the reduced temperature variable $t = (T - T^*)/T^*$. The blue triangles were obtained from a sequence of images similar to the one shown in Fig. 6a–f but recorded while heating the sample. Note that at lower temperatures the $L(T)$ values obtained on heating and cooling separate out clearly, indicating the development of an out-of-equilibrium situation towards low temperatures (discussed in ref. 40). This may result from quenched disorder, for example, due to local fluctuations of the Fe thickness that hinders the motion of domain walls and thus prevents the system from adjusting $L$ to its equilibrium value[44]. This out-of-equilibrium situation may also have an intrinsic origin as a manifestation of glassiness[47], which has been predicted for the model Hamiltonian (1)[14,15,20,22]. (We recall that the $M(T, B)$ data used to produce scaling plots and the phase diagram are, instead, reasonably equilibrated since they are independent of the sweeping rate of $B$.) Close to $T^*$ ($t = 0$) the values of $L(T)$ merge within the experimental error.

The green and red crosses in the main frame of Fig. 6g and the coloured points in the inset were not obtained by imaging—for the simple reason that they extend above $t = 0$, where mobility makes stripe invisible to SEMPA imaging[45]. These additional, essential data on $L(T)$ were deduced from an equation that relates $L(T)$ to the spatially averaged magnetization $\mathcal{M}(T, B)$ in small applied magnetic fields[38,39] (see the subsection 'Experimental determination of $L(T)$ using $\mathcal{M}(T,B)$' in Methods):

$$L(T) = \mu_0 \frac{\pi}{4} d \left[ \frac{\partial \mathcal{M}(T, B)}{\partial B} \right]_{B=0} \quad (4)$$

$d$ being the thickness of the film. We point out that in equation (4) $\mathcal{M}$ is the spatially averaged magnetization expressed in physical units (to be distinguished from $M$, which is the

spatially averaged magnetization measured by MOKE and is in arbitrary units). The response of $\mathcal{M}$ to $B$ is well defined both below and above $t = 0$, see, for example, Fig. 6h: accordingly, $\mathcal{M}(B)$ can be used to determine $L(T)$ via equation (4) in the range of temperatures where stripes are no longer observed in SEMPA. Figure 6g shows that below $t = 0$, where both imaging (blue triangles and pink squares) and $\mathcal{M}(T, B)$ (green and red crosses) can be used to deduce the stripe width, SEMPA and equation (4) give almost the same values for $L(T)$. This correspondence strongly supports the validity of equation (4) itself. Remarkably, the data points represented by green and red crosses (and the coloured ones in the inset) continue into the temperature region where SEMPA imaging becomes contrastless, $t > 0$. This suggests that the stripe width, directly observed on static patterns below $t = 0$, evolves smoothly up to the highest measured temperatures, $t \approx + 0.16$ corresponding to more than 30° above $T^*$ (inset of Fig. 6g), and ranges from several micrometres at low $T$ down to few tens of nanometres[32]. The estimates of $L(T)$ represented with crosses, triangles and squares refer to a slightly thicker Fe film (2 MLs) than the one (1.75 MLs) on which MOKE measurements and scaling analysis were performed. The Curie temperature determined in the thinner film as described before is indicated in the inset as $t_c$. Figure 6g thus establishes a characteristic long spatial scale $L$ at every temperature, carrying neither a sign of the transition to mobile stripes (at $T^*$) nor an anomaly at the putative critical temperature $T_c$. Equation (4) establishes a proportionality between $L(T)$ and the magnetic susceptibility $\mu_0 \cdot \left[ \frac{\partial \mathcal{M}(T, B)}{\partial B} \right]_{B=0}$. As $L(T)$ is a non-singular quantity as a function of the temperature, we expect the susceptibility as well to be a non-singular quantity as a function of the temperature within the grey zone. Indeed, the nonlinear relationship between $M$ and $B$, detected outside the grey zone, gives way to the linear, non-critical behaviour shown

in Fig. 6h (the susceptibility being the slope of the graphs in Fig. 6h at low magnetic fields). This remarkable difference has been anticipated in the sketches in Fig. 1a,d. From magnetic imaging we can associate the linear portion of the $M(T, B)$ curve in Fig. 6h with the displacement of domain walls, which accompanies the increase of those magnetic domains whose magnetization is parallel to $B$ at the expense of the size of domains with magnetization antiparallel to the magnetic field[39].

A legitimate question is whether the period of modulation actually acts as a natural cutoff that hinders the divergence of the correlation length $\xi$ underlying the critical behaviour in the unfrustrated system. (We thank the reviewers for putting these issues under our attention (see Reports).) Should this be the case, then finite-size-scaling relations[54,55], involving both $\xi$ and $L$, should replace the equations of state of the 2D-Ising ferromagnet. In the subsection 'Scaling and characteristic lengths' in Methods we show that finite-size-scaling relations are incompatible with the law $\mathcal{M} \sim L \cdot B$ observed experimentally in the patterned phase (inset of Fig. 6g) and provide an additional argument on this issue.

## Discussion

With the plots of Figs 2b, 4b,c and 5 we have discovered a strikingly simple but, surprisingly, yet unnoticed symmetry: even if an indefinitely weak long-ranged and frustrating interaction—the dipolar coupling in our case—is enough to eliminate the critical point of a standard second-order phase transition, sufficiently away from it ordinary critical scaling à la Kadanoff is fully recovered[3–6]. In our specific Fe films, outside the region of the phase diagram corresponding to the patterned phase (grey zone) we observe 2D-Ising scaling, despite the fact that the true critical point at $(T, B) = (T_c, 0)$ is—strictly speaking—avoided. It remains to be determined what type of order is realized within the grey zone and how it relates to the physics outside of it, namely in the ferromagnetic scaling region. As already anticipated, strong theoretical arguments exclude the persistence of positional order of the stripe domain pattern at finite temperature[21]. For what concerns orientational order, the lowest-energy perturbations can be described by an orientational effective Hamiltonian that explicitly possesses O(2) symmetry[21] (symmetric with respect to continuous rotations around a fixed axis). Both these facts are compatible with the occurrence of a Berezinskii–Kosterlitz–Thouless phase transition from a nematic phase of magnetic domains to a disordered phase[17,18,21] and our films might, indeed, be good candidates to observe such a transition. The mixing of $\mathbb{Z}_2$ and O(2) symmetries is encountered in other models as well, like the 2D frustrated XY models[56–61]. In the last ones, the O(2) symmetry is explicit in the Hamiltonian, while the $\mathbb{Z}_2$ symmetry emerges from the chiral degree of freedom of the ground state; the interplay between the two symmetries gives rise to a rich—and debated—scenario of phase transitions[57–61]. However, the patterned phase of the model defined by equation (1) is expected to evolve differently with temperature with respect to frustrated XY models: in the model of our interest the $\mathbb{Z}_2$ symmetry—broken in one ground state—is restored at finite temperature by elastic-like excitations, while in genuine Ising-like transitions it is restored creating domain-wall excitations. These important open issues will certainly stimulate further work to better characterize the grey zone. From the present investigation we definitely conclude that whatever order is realized in the modulated phase of thin films magnetized out of plane, it must be such that (i) standard scaling relations, like $M/|\tau|^\beta = g_\pm(B/|\tau|^{\beta\delta})$ or $B/M^\delta = f(\tau/M^{1/\beta})$, are replaced by the one-variable relation $M/B \sim L(T)$ (In ref. 28 this scaling law is generalized to any dimension and to a generic power-law

exponent $\alpha$ in equation (1).) and (ii) a well-defined modulation length $L(T)$ persists deep into the paramagnetic phase.

## Methods

**Technical details of Monte Carlo simulations.** Data in Fig. 2a–c were produced by Monte Carlo simulations on a square lattice with $L_x = L_y = 120$ using the Hamiltonian (1). Some simulations were also run for $L_x = L_y = 200$ to exclude possible finite-size effects and to produce the snapshots in Fig. 2c. Ewald sums technique was used to implement the long-range interaction with $\alpha = 3$ for periodic boundary conditions[27,62]. This term represents the dipolar interaction in the Ising limit with the easy axis perpendicular to the film plane, in which case dipole–dipole coupling is antiferromagnetic for all spin pairs; the corresponding sum runs over all the $(i,j)$ pairs of distinct sites on the lattice. The limiting field $b_c$ above which the system is in a phase with mostly uniform spin profile was determined from the behaviour of the average magnetization $m$ as a function of temperature for fixed values of $b$, according to a zero-field-cooled–field-cooled protocol. After having prepared the system in a uniformly saturated state (with $m = 1$), we let the temperature increase from very low values up to a predefined $T_f$, lying above the transition from the modulated to the paramagnetic phase for $b = 0$. Then, we stopped the simulation and, starting from the final configuration, the system was cooled down to the original temperature. When $b < b_c$ a strong hysteresis in the $m(T)$ curves was observed, below the temperature where modulated phases develop. When $b > b_c$, instead, the curves were completely reversible in the whole temperature range. In this way we could estimate $b_c$ for rather large values of $J/g$ (up to $J/g = 10$), without the computational cost of complete phase-diagram calculations[27]. To observe domain phases within the given $L_x$ the ratio $J/g$ cannot be as large as in the experiment, where $T_c/g \sim O(10^2)$, because the width of domains increases exponentially with this ratio[27,28], and realistic simulations can be performed up to about $L_x = L_y = 200$ (the necessity of summing over all pairs, imposed by the dipolar interaction, limits the size of the simulation box significantly with respect to the cans in which only short-ranged interactions are considered).

**SEMPA imaging.** Spatially resolved magnetic imaging shown in Fig. 6a–f is performed with SEMPA (Fig. 3b). A focused electron beam of primary electrons is directed towards the sample and the secondary electrons excited off the topmost surface layers by the primary electrons are sampled and analysed according to their spin polarization, which is proportional to the local magnetization vector within the beam focus. The spin polarization is rendered in the images by a grey scale, black and white corresponding to opposite spin polarization. Only the component perpendicular to the surface is displayed, as the components of the spin polarization parallel to the film plane are vanishing. The images are built by horizontal line scans, consisting of about 200 pixels, from top to bottom, starting from the left-hand side of each image. For contrast to be detected, the magnetization distribution must be static over the times required to collect at least a dozen horizontal lines, that is, about 60 s.

**Experimental determination of $T_c$.** In conventional ferromagnets one finds $T_c$ as the critical temperature at which, for example, the critical isochamp at zero applied magnetic field $M(T, B = 0)$ vanishes. In the present system, the ordinary critical point is obscured, at low magnetic fields, by the appearance of static and mobile domains, so that one has to find a different way to determine $T_c$. One useful property of a conventional second-order phase transition is that the plot of the magnetic susceptibility $\chi(T, B) = \mu_0 \cdot \partial \mathcal{M}(T,B)/\partial B$ as a function of temperature has a maximum at a temperature $T_{max}(B)$ that approaches $T_c$ as $B$ approaches zero[63]. Figure 7a shows the susceptibility $\chi(T, b)$ of the unfrustrated (that is, without dipolar interactions) 2D-Ising model computed by means of Monte Carlo simulations. For any $b \neq 0$ the maximal value of $\chi(T, b)$ is finite but—as expected—it is higher the weaker the fields. Notice that the true approach of $T_{max}(b)$ to $T_c$ may not necessarily be linear in $b$, as pointed out in ref. 63, but our numerical simulations (Fig. 7b) show that a linear extrapolation of $T_{max}$ towards $b = 0$ gives a fairly accurate estimate of $T_c$.

Experimentally, we obtained the $\chi(T, B)$ by first treating the raw $M(T, B)$ data with a Savitzky–Golay finite-impulse-response smoothing filter implemented in MATLAB. After filtering, the derivative $\chi(T, B) = \mu_0 \cdot \partial M(T, B)/\partial B$ could readily be obtained. In Fig. 7c some of the resulting susceptibility curves are shown for selected values of $B$. The experimental $B(T_{max})$ is plotted in Fig. 7d. This graph, in contrast to the one in Fig. 7b, shows two distinct regimes, depending on the range of $B$. We discuss first the low field regime. When cooling in weak enough fields, the system first enters the grey zone, (see point '1' in Fig. 3a), without displaying any anomaly in the susceptibility in correspondence with this transition. On further cooling a re-entrant transition (point '2' in Fig. 3a) from the patterned to the uniform phase is encountered[39]: this second transition is accompanied by a sharp maximum in the susceptibility (abrupt increase of the magnetization). This type of maxima—highlighted by a blue dashed line in Fig. 7d—mark the transition from a phase with domains to the uniform phase. When extrapolated to $B = 0$, they lead to a temperature at which the sample consists of very large stripes carrying opposite but almost saturated values of the magnetization. Above this temperature, the sample keeps the modulated order up $T^*$, where domains become mobile but the

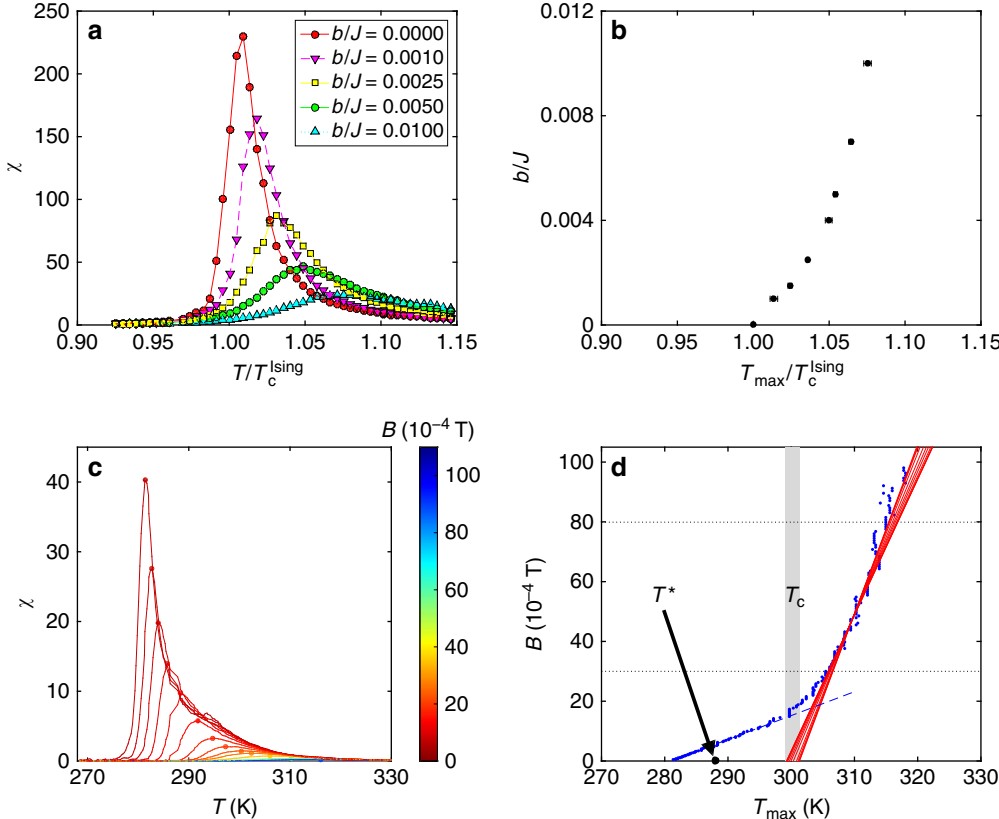

**Figure 7 | Determination of the Curie temperature** (**a**) Plot of the magnetic susceptibility $\chi(T,B) = \partial m/\partial b$ ($b$ in units of $J$) a function of $T/T_c$ for a lattice of linear size $L_x = L_y = 120$, from Monte Carlo simulations for the unfrustrated 2D-Ising model obtained setting $g = 0$. We recall that $T_c^{Ising} = 2.269J$, assuming $k_B = 1$; the selected values of $b/J$ are given in the figure legend. (**b**) $b/J$ versus $T_{max}(B)/T_c^{Ising}$; for each field, the value of $T_{max}(B)$ was extrapolated with FSS using lattices of size $L_x = L_y = 48$, 64, 96, 128 and 200. (**c**) Plot of the magnetic susceptibility $\chi(T,B) = \mu_0 \cdot \partial M(T,B)/\partial B$ of a 1.75-ML-thick Fe/Cu(001) film as a function of temperature for different magnetic fields. Different colours correspond to different applied fields (see colour map on the right). (**d**) Plot of $B$ versus $T_{max}(B)$ obtained from the experimental susceptibility as described in the subsection 'Experimental determination of $T_c$' in Methods.

magnetization within them is still substantial. Accordingly, $T_c$ must be above $T^*$. On the other side, for larger fields (the right-hand side portion of the graph) the grey zone of Fig. 3a is never entered and the system is in a uniformly magnetized phase: $B(T_{max})$ for larger $B$ was therefore taken to extrapolate towards a putative $T_c$. Several linear fittings were performed by choosing different ranges of the field $B$ between $30 \times 10^{-4}$ and $80 \times 10^{-4}$ T. The different fittings produced the family of red lines (Fig. 7d) from which the error on the $T_c$ was estimated. The experimental $T_c$, for this particular sample, is $300 \pm 1$ K. Notice that the extrapolated $T_c$ lies within the cross-over range (K) $298 \lesssim T \lesssim 302$, in which the 'blue-line' types of maxima transform into the 'red-line' types of maxima.

**Experimental determination of $\beta$ and $\delta$.** In conventional ferromagnets one finds the critical exponent $\beta$ via the asymptotic behaviour of the critical isochamp near $T_c$, that is, from $M(T, B = 0) \sim (-\tau)^{\beta}$ and the critical exponent $\delta$ via the asymptotic behaviour of the critical isotherm, that is, $M(T_c,B) \sim B^{1/\delta}$. In the present system, these ordinary power laws are obscured, at low magnetic fields, by the appearance of static and mobile domains. However, the conventional lore of scaling (ref. 1, p. 485) and our simulations for the specific system, indicate practical ways to find $\beta$ and $\delta$ using experimental MOKE data of $M(T, B)$ originating within the high-temperature, non-zero field region of the $(T, B)$ parameter space. The critical exponent $\beta$ can be deduced from the experimentally determined values of the exponents $\gamma$ and $\delta$, using the relation $\beta\delta = \beta + \gamma$. The exponent $\gamma$ determines the magnetization in the region of weak fields (equation (148.8) in ref. 1) according to

$$M \sim \frac{B}{|\tau|^{\gamma}} \qquad (5)$$

The exponent $\delta$ determines the magnetization in the region of strong fields according to the relation

$$M \sim B^{1/\delta} \qquad (6)$$

(see equation (148.10) in ref. 1). The notion of weak and strong field is, of course, dependent on which temperature interval is addressed. In Fig. 8a we plot $\log(M/B)$ versus $\log|\tau|$ for different values of $B$ (for the colour code indicating the values of $B$ see the horizontal scale in the inset). At sufficiently high temperatures, a region of the graph emerges, where all the curves for different magnetic fields

almost collapse onto one single straight line, and thus fulfil the scaling properties required by equation (5) for the quantity $M/B$. The negative of the slope of the resulting straight line is the sought-for exponent $\gamma$. Several linear fittings were performed for fixed fields ranging from $20 \times 10^{-4}$ to $80 \times 10^{-4}$ T. These independent determinations of $\gamma$ are shown in the inset. After averaging, for $T_c = 300$ K we obtain $\gamma = 1.78 \pm 0.09$, with the error given by the s.d. of the mean. As consistency check, the whole procedure was repeated varying the value of $T_c$. The s.d. of the mean values of $\gamma$ goes through a minimum in the range (K) $298 \leq T_c \leq 301$, which is, accordingly, the range where the 'best collapsing' of the $M/B$ curves is realized. When $T_c$ is varied in this interval, $\gamma$ ranges from 1.7 to 1.9.

The $\log(M/B)$ plot of Fig. 8b in the temperature range (K) $299 < T < 301$, reveals a low-field region where the curves saturate to an almost constant value, indicating the linearity of $M$ versus $B$ for weak fields. In the strong-field region the graphs are observed to almost collapse onto a single straight line, the slope of which amounts to $(1 - \delta)/\delta$, consistently with equation (6). From these slopes fitted for different $T$ in the appropriate regime of Fig. 8b we estimate $\delta = 13 \pm 2$.

**Scaling and characteristic lengths.** As suggested by one of the reviewers, in a conventional ferromagnet scaling plots can also be written in such a way that the dependence on the correlation length $\xi \sim |\tau|^{-\nu}$ is made explicit ($\nu$ being the corresponding critical exponent). Let us consider the following equations of state of a ferromagnet:

$$\frac{M(T,B)}{B^{1/\delta}} = F_1\left(\frac{B^{-\nu/(\beta\delta)}}{|\tau|^{-\nu}}\right) \qquad (7)$$

and

$$\frac{M(T,B)}{|\tau|^{\beta}} = F_2\left(\frac{B^{-\nu/(\beta\delta)}}{|\tau|^{-\nu}}\right) \qquad (8)$$

with $F_1(x)$ and $F_2(x)$ some scaling functions. In Fig. 5b,c the same data points shown in Fig. 4c are plotted in the representations defined above. Points falling in the grey zone are rendered in grey. Indeed, collapsing is observed for the coloured points while the grey ones spread out in the plot planes. Figure 5c appears

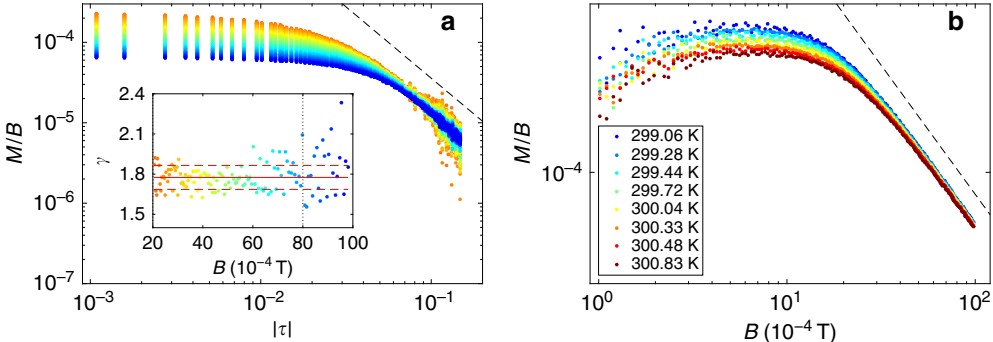

**Figure 8 | Determination of the exponents $\beta$ and $\delta$.** (**a**) Log–log plot of the ratio $M/B$ versus $\tau = (T - T_c)/T_c$ (with $T_c = 300$ K). The dashed black line is a guide to the eye with slope equal to the mean value of $\gamma$. Different isochamps are plotted with different colours: specific values of the field $B$ can be identified from the horizontal axis of the inset. The inset plots the fitted values of $\gamma$ as a function of the magnetic field. The solid and dashed horizontal lines indicate the mean value of $\gamma$ and the s.d. from this average, respectively. The vertical dotted line marks the largest field $B = 80 \times 10^{-4}$ T used to determine $\gamma$. (**b**) Log–log plot of the ratio $M/B$ versus $B$. Different isotherms are plotted with different colours (specific values of $T$ are given in the legend). The dashed black line is a guide to the eye with slope equal to mean value of $(1 - \delta)/\delta$ (fitted exponents).

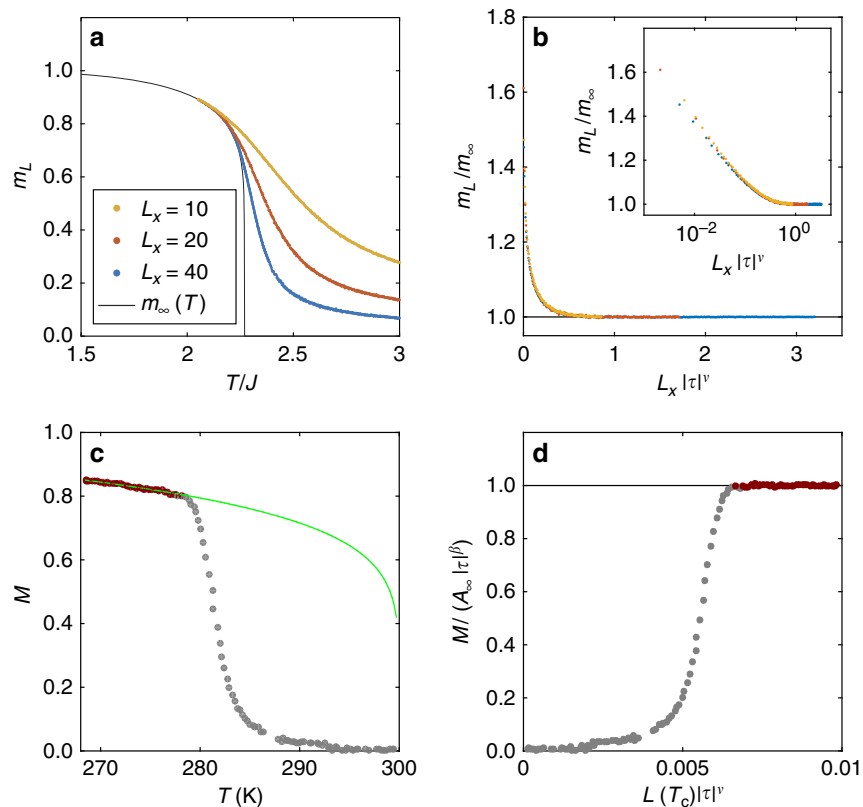

**Figure 9 | Test of FSS.** (**a**) Monte Carlo simulations of $m_L(T)$ versus $T$ computed for the unfrustrated 2D-Ising model and $b = 0$ for a square lattice of linear size $L_x = 10, 20, 40$ (colour code in the legend). The black curve is the exact Onsager solution $m_\infty(T)$. (**b**) Theoretical FSS function $f_M(x)$ deduced from the data in **a** as described in the subsection 'Scaling and characteristic lengths' in Methods. Inset: same data plotted in a log–linear scale. (**c**) Experimental $M(T)$ data versus $T$ measured for $B = 5 \times 10^{-5}$ T. The green line corresponds to $M_\infty = A_\infty |\tau|^\beta$, with $\beta = 1/8$ and $A_\infty = 1.192 \times 10^{-2}$. (**d**) Ratio $M_L(T)/|\tau|^\beta$ versus $L(T_c) \cdot |\tau|^\nu$ representing the function $F_4(x)$ defined in equation (14).

very similar to Fig. 4c and basically the same considerations apply. Equation (7) leads instead to the plot in Fig. 5b that has a different shape.

If the spreading of grey points observed in all the scaling plots was due to the size of magnetic domains $L$ acting as a cutoff for the correlation length, from equation (7) one would expect the following relation to be fulfilled in the patterned phase

$$\frac{M(T,B)}{B^{1/\delta}} = F_3\left(\frac{B^{-\nu/(\beta\delta)}}{L}\right) = \tilde{F}_3\left(B^{\nu/(\beta\delta)} \cdot L\right) \quad (9)$$

with $F_3(x)$ and $\tilde{F}_3(x)$ appropriate scaling functions. This scaling relation is not compatible with the one found in within the grey zone, $M_{\mathrm{grey}}(T,B) \sim B \cdot L$, which

can be rewritten as

$$\frac{M_{\mathrm{grey}}(T,B)}{B^{1/\delta}} \sim B^{(\delta-1)/\delta} \cdot L \quad (10)$$

Equations (9) and (10) can be simultaneously fulfilled only for

$$\delta - 1 = \frac{\nu}{\beta} \quad (11)$$

This last relation is not obeyed by the 2D-Ising critical exponents $\beta = 1/8$, $\delta = 15$ and $\nu = 1$. A more proper framework to discuss whether $L$ acts as a cutoff for the correlation length is provided by the finite-size scaling (FSS) ansatz[54,55], according

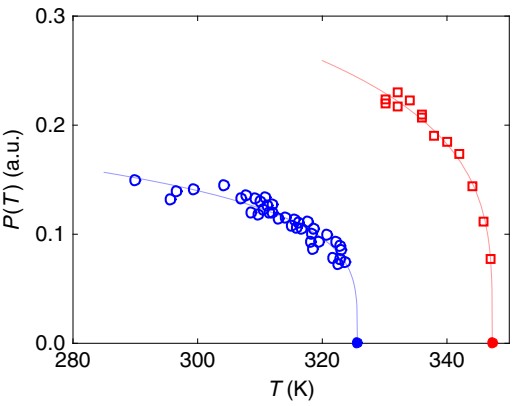

**Figure 10 | Local magnetization as a function of temperature.** The local magnetization, given by the spin polarization $P$ of the secondary electrons, is extracted from SEMPA images, adapted from ref. 39 (red) and ref. 45 (blue). Solid lines are guides to the eye obtained with a phenomenological power-law fitting $\sim (T^*-T)\beta_{eff}$ that yields an effective $\beta_{eff} \simeq 0.25$ in both cases. Note that the two samples have different temperatures $T^* = 326$ and 347.2 K, resulting from the fit and indicated by full dots.

to which the magnetization $M_L(T)$ of a finite system of linear size $L_x$ should be related to the magnetization of the corresponding infinite systems $M_\infty(T)$ by the law

$$M_L(T) = M_\infty(T)f_M(L_x/\xi) \tag{12}$$

where $B = 0$ is assumed and $f_M(x)$ is some FSS function of the single variable $x = L_x/\xi$ such that

$$\begin{aligned} f_M(x) \rightarrow 1 &\quad \text{for } L_x \gg \xi \\ f_M(x) \rightarrow \text{const.} &\quad \text{for } L_x \ll \xi \end{aligned} \tag{13}$$

Figure 9a displays the $m_L(T)$ as a function of $T$ computed for the unfrustated 2D-Ising model and $B = 0$ for finite square lattices of size $L_x = 10, 20, 40$. Comparatively small lattices were chosen to enhance finite-size effects. The exact Onsager solution for the infinite system, $m_\infty(T)$, is also shown as a solid black line. From these data the theoretical FSS function $f_M(x)$ can be deduced simply by plotting $m_L/m_\infty$ versus $L_x|\tau|^\nu$, the latter being proportional to $L_x/\xi$ (Fig. 9b). The same procedure was repeated for the experimental points assuming $M_\infty = A_\infty|\tau|^\beta$, with $\beta = 1/8$ and $A_\infty$ fitting parameter. Figure 9c demonstrates that this law is followed by the coloured points, while the grey points deviate from it. A speculative FSS function, in which the role of the lattice size $L_x$ is played by the size of magnetic domains at the putative Curie temperature $L(T_c)$, can be obtained directly from experimental points and reads

$$\frac{M(T, B=0)}{|\tau^\beta|} = F_4(L(T_c)\cdot|\tau|^\nu) \tag{14}$$

The function $F_4(x)$ is plotted in Fig. 9d. The part of curve resulting from grey points deviates from $f_M(x)$ computed for the 2D-Ising model shown in Fig. 9b, which seems to confirm that $L$ does not simply act as a cutoff for the correlation length $\xi$.

**Magnetization within domains.** Below $T^*$ domains are frozen and one can reliably extract from SEMPA images, like the ones shown in Figs 2d and 6a–f, the value of the local magnetization, that is, the magnetization within the domains. The data related to this quantity, published in two previous articles[39,45], are reported in Fig. 10 for convenience. As anticipated in the main text, the local magnetization inside the stripe domains decreases sizeably while $T^*$ is approached. If one tries to capture this decrease using a power law, one obtains a curve, which vanishes in the vicinity of $T^*$, with an effective critical exponent of about 0.25 (Fig. 10). However, from the present work we know that the putative $T_c$ deduced from scaling analysis lies about 30° above $T^*$. Therefore, the power-law behaviour represented by the continuous curves in Fig. 10 is not related to the scaling behaviour observed in the ferromagnetic scaling region. We point out that we are not aware of any prediction that assigns a power law to the local magnetization in the vicinity of $T^*$. Moreover, we note that our experimental data in Fig. 10, strictly speaking, do not necessarily speak for a power-law vanishing of the local magnetization.

**Experimental determination of $L(T)$ using $\mathscr{M}(T, B)$.** In the temperature region $t < 0$ stripe domains are static and can be imaged with SEMPA. By means of equation (4) the stripe width can also be determined from the behaviour of the spatially averaged magnetization $\mathscr{M}(T, B)$ at low magnetic fields. $\mathscr{M}$ is the product of $|\mathscr{M}_0|$—the value of the local magnetization within the stripes—and $\mathcal{A}(T, H) \doteq (f_\uparrow - f_\downarrow)/(f_\uparrow + f_\downarrow)$, that is, the asymmetry between the film

area occupied by up ($f_\uparrow$), respectively, down ($f_\downarrow$) perpendicular magnetization: $\mathscr{M}(T, B) = \mathscr{M}_0(T, B) \cdot \mathcal{A}(T, B)$. In the range $t < 0$ and for small $B$, $\mathscr{M}_0(T, B)$ is a smooth function of $T$ and almost independent of $B$. $\mathcal{A}(T, B)$, instead, increases, close to $T^*$, linearly[39] with $B$ and describes a process where the width of stripes magnetized parallel to $B$ increases at the expenses of the width of stripes magnetized antiparallel to $B$. Accordingly, the following scaling law has been demonstrated[39]: $\mathscr{M}(T, B)/B \propto L(T)$, that is, the response of $\mathscr{M}$ to $B$ is linear (Fig. 6h) and the susceptibility is proportional to the sought-for equilibrium stripe width. The knowledge of the proportionality constant is thus crucial to determine $L(T)$. This can be computed exactly for a square-like stripe profile (see equation (2.38) in ref. 38), which yields equation (4). A perfect square profile is not expected at finite $T$ because the domain walls are certainly not atomically sharp. However, as shown in ref. 39, the assumption of an almost square profile explains precisely the response of $\mathcal{A}$ to $B$ also in the very vicinity of $t = 0$, so that we are confident that the proportionality constant computed for a perfect square profile holds up to temperatures where $L(T)$ behaves smoothly (inset, Fig. 6g). Note that $\mathscr{M}$ entering equation (4) is expressed in physical units, while in MOKE measurement we access $M$, which is in arbitrary units. For low temperatures and sufficiently large applied fields the film is in the uniform phase with all magnetic moments almost fully aligned, so that the value of the MOKE signal $M_S$ measured in this condition can be associated with the saturation magnetization of Fe, $\mu_0 \cdot \mathscr{M}_S = 2.16$ T. The other measured values were rescaled accordingly to obtain the physical magnetization $\mathscr{M}(T, B)$.

**Data availability.** Data are available from the ETH Zurich Data Archive: http://doi.org/10.5905/ethz-1007-20.

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

## Acknowledgements

We thank Thomas Bähler for technical assistance, G. M. Graf, A. Giuliani, O. V. Billoni and S. Ruffo for helpful discussions, as well as the Swiss National Science Foundation, ETH Zurich and CONICET (Argentina) for financial support. D.P. dedicates the paper to V. Pokrovsky, on the occasion of his 85th birthday, for introducing to him the subject of low-dimensional physics.

## Author contributions

N.S. developed the instruments, conceived the experiment, collected the experimental data and discussed the results; D.A.Z. developed the software for data analysis, analysed the data and discussed results; U.R. developed the instruments, collected the experimental data, discussed the results and supervised the experimental work; S.A.C. did the Monte Carlo simulations and discussed the paper; D.P. discussed the results and wrote the paper; A.V. conceived the experiment, analysed the data, discussed the results, wrote the paper and supervised research.

## Additional information

**Competing financial interests:** The authors declare no competing financial interests.

DOI: 10.1038/ncomms14372    **OPEN**

# Corrigendum: Critical exponents and scaling invariance in the absence of a critical point

N. Saratz, D.A. Zanin, U. Ramsperger, S.A. Cannas, D. Pescia & A. Vindigni

*Nature Communications* 7:13611 doi: 10.1038/ncomms13611 (2016); Published 5 Dec 2016; Updated 17 Jan 2017

The original version of this Article contained a typographical error in the spelling of the author S.A. Cannas, which was incorrectly given as S. Cannas. This has now been corrected in both the PDF and HTML versions of the Article.

