## [Peer Review File · Nature Communications]

Reviewers' comments:

Reviewer #1 (Remarks to the Author):

REPORT:

The authors of this paper studied an interesting question concerning the scaling invariance and critical exponents of a phase transition in a case called "avoided critical point".

This question may attract deeper investigations in statistical physics to extend the picture of a phase transition provided by the renormalization group to more complicated and realistic materials.

To their purpose, the authors used experimental data they performed on a monolayer (in fact 1.7 ML) of Fe using SEMPA and MOKE techniques which provided them with a phase diagram in the (T,B) plane where T is the temperature and B the applied magnetic field. The diagram is shown in Fig. 1 (and also in Fig. 2d). They stated that the spin of the layer are perpendicular to the film plane with a ferromagnetic interaction between nearest neighbors and a infinite-range dipolar interaction. They argued that the latter exists in the Fe film (they did not explain why but I think this is from their earlier experimental works). This polar interaction is at the origin of the modulated phase a in region around the would-be critical point T_c if this modulated phase does not exist. They deduced T_c which fits the scaling laws with a large number of experimental data (but not all). They found $T_c=300$ K and critical exponents $\beta=0.15 \pm 0.03$ and $\delta=13 \pm 2$. Within errors these exponents are those of the exactly solved 2D Ising model ($\beta=0.125$, $\delta=15$). The data which do not fit the scaling law are scaled with the so-called one-variable scaling relation with g functions as shown in Fig. 2c with the 2D Ising critical exponents. The remaining data which do not collapse even with this law were interpreted as those belonging to the grey zone in Fig. 2d. These points that violate 2D-Ising scaling, whose number is substantial, are assigned to a phase with modulated magnetization (grey zone). In this modulated phase, the scaling is $M/B \propto L(T)$ (width of modulated domains) as suggested in the paper PRL (2010) from the authors.

They have also simulated the system using the Monte Carlo method with the Hamiltonian (2). Using a slow heating and a slow cooling in a field, they observed a "critical" value of the field b_c below which the modulated phase sets in. The value of T_c depends on the dipolar strength. The fit with the 2D Ising scaling law is observed for $b > b_c$ where the magnetization is uniform.

The above description is what I see from what the authors stated in the manuscript. Here are my remarks:

1. On the presentation: The manuscript is difficult to read. This is because the authors go back and forth on the scenario at many places which are not easy to follow. It would be preferable that they anticipate the scenario once at the beginning of section II and show the results to justify that. I appreciate on the other hand the section Methods which is well presented.

2. On the physics:

a) The authors adjusted the value of the critical temperature T_c in order to realize the best collapsing of data on the 2D Ising scaling curve. To my opinion, there are two possibilities:

(i) The first is that nothing can warrant that such a T_c exists or corresponds to a physical reality because no experimental data in this paper show a sign of a phase transition,

(ii) The second possibility is that T_c does really exist though no signature has been given here. The fact that scaling law works using its value means that there may be a kind of mixing of Ising criticality

and non critical fluctuations due to modulated structure. The critical point is not "avoided" but masked by another kind of fluctuations. We have seen in the literature that the mixing of various kinds of symmetries is possible (for example in fully frustrated XY spin systems such as the Villain's domino model or the antiferromagnet triangular lattice, there is a mixing of vortex (KT phase) and the two-fold (Ising) chirality, the transition can be fit with slightly modified Ising critical exponents in spite of the KT nature => there was no need to search for a fictive T_c). In the case studied here, the modulated structure can coexist with an Ising long-range correlation (it can be a kind of staggered magnetization defined using the period of the modulation width at T_c which is experimentally known, namely $L(T_c)$). So the wording in the title and in the text may be misleading if the second possibility is a reality.

b) The Hamiltonian is for an Ising model with spins pointing in the direction perpendicular to the film thickness. This assumption eliminates the second term in the dipolar interaction, retaining only the antiferromagnetic term. My question is: what is the value of the crystalline Fe spin ? is there any reason to suppose that the Fe spins are of the Ising type? Is there any anisotropy along the z direction to justify such an assumption?

c) In the phase space (T,B) where the spin configuration is ferromagnetic with B not zero (the blue horizontal line in Fig. 1), we can't talk about "critical" region because , strictly speaking, ferromagnets under an applied field does not make a phase transition.

d) On the simulation: The dipolar term is not a simple spatial "lattice sum" because each term of the sum depends on the instantaneous value of each pair $S_i.S_j$ (+1 or -1) during the simulation. For each spin pair, this value changes with time-evolution by thermal fluctuations. Can the authors explain how the Ewald's sum is performed taking into account such a real instantaneous spin configuration at each MC sweep? Also, when J/g is very large, the dipolar effect decreases, the spin configuration should tend to ferromagnetic. Thus, the interesting region is that of small J/g .

e) I suggest the authors to add a snapshot of the spin configuration in the modulated region to show the dipolar effect.

In conclusion, while the subject is interesting I found that a number of points should be clarified before reconsidering the manuscript for publication.

Reviewer #2 (Remarks to the Author):

The authors study and discuss the (T,B) phase diagram of Fe films. This phase diagram presents a paramagnetic phase, a ferromagnetic phase, and a modulated domain phase, induced by the dipole-dipole interaction. The authors report critical fluctuations associated to the paramagnetic-ferromagnetic transition over many decades of the scaled variables in spite that the corresponding Curie point is actually avoided, as they explicitly show. They compare their results with numerical predictions of a 2d-Ising model with dipolar interactions.

Although the phenomenon of avoided criticality has been described in previous works, as discussed in the introduction of the manuscript, it has not been considered before for this "model" experimental system. I consider the experimental results are impressive and the analysis is simple and rather convincing. In addition, the quality of the presentation is good: the main message of the paper is

simple and clear cut, and of general interest. Understanding the details will surely motivate further research.

I would like to recommend this paper for publication in Nature Communications, after the authors address the following questions/suggestions.

1)

The critical scalings can also be conveniently written in terms of characteristic lengths, by using the corresponding ν exponent:

$$M(T,B) \sim b^{1/\delta} F_1(B^{-\nu/\beta} \delta / \tau^{-\nu})$$

or

$$M(T,B) \sim \tau^{\beta} F_2(B^{-\nu/\beta} \delta / \tau^{-\nu})$$

I wonder if the authors have tried to check for the following relations (latex formulas):

$$M(T=T_c, B) \sim B^{1/\delta} F_3(B^{-\nu/\beta} \delta / L(T_c))$$

and

$$M(T, B=0) \sim \tau^{\beta} F_4(\tau^{-\nu} / L(T))$$

when we approach the modulated phase at $T=T_c$ or $B=0$ respectively, and where $L(T)$ is the period of the modulated phase. (F_1, F_2, F_3, F_4 are just scaling functions. I am omitting the \pm differentiation for simplicity and I mean $\tau = |T - T_c| / T_c$).

In other words, my question is whether $L(T)$ just acts as a cut-off for the paramagnetic-ferromagnetic divergent correlation length (or susceptibility), inducing the above scalings. Please comment.

2) Please comment about the disorder in the samples. Does it have any kind of impact on the obtained results?.

3) Is there a theoretical prediction for the (B, T) phase diagram to compare with the experimental one?. If so, does the model of Eq(2) contain the minimal ingredients to describe it?. Please comment.

4) There is a "long-range" typo in the text.

REVIEWERS' COMMENTS:

Reviewer #1 (Remarks to the Author):

In the revised version, the authors have taken into account all the remarks raised in my first report. I read the replies of the authors point by point. They agreed with my remarks and have made a great effort to satisfy all of them in the revised manuscript: in particular the new presentation of the paper(theory first, experiments next), the wording at some delicate points, adding a comparison with the frustrated XY model, adding some MC snapshots to compare with the SEMPA image, ...

I think that as far as my remarks are concerned, the revised paper can be published in Nature Communications.

Reviewer #2 (Remarks to the Author):

I consider the authors have satisfactorily answered the questions, and presented a rather improved version of the manuscript.

Unfortunately however, I can not see the new Fig S7 in the supplement. I'll be ready to make my recommendation after seeing it.

RESPONSE TO THE REVIEWERS

We would like to thank both Reviewers for the constructive criticism and useful suggestions They provided, which obviously result from a careful review.

As a results of the review process, we believe that the manuscript is now improved and conveys the main outcomes of our work more straightly than previously.

A list of the major changes that were made followed by a point-by-point response (blue) to the Reviewers' comments (black) are provided below.

Major changes

- Apart from re-arrangements and different numbering, we changed Figure 1,2,4b in order to comply with the requests of the Reviewers.
- New Monte-Carlo simulations were performed to produce the snapshots in Fig.2c and the corresponding curve in Fig.2a (for $b/g = 0.1$).
- New experimental images (Fig.2d) were added to compare with microscopic Monte-Carlo configurations (snapshots).
- Fig.S5, S6, S7, S8 were added to address the points raised by the Reviewers (including new Monte-Carlo simulations, Figs.S7a,b).
- The section “Theoretical scenario” was enlarged and anticipated with respect to the description of experiments (see the answer to the specific comment of the Reviewer #1).
- We introduced the notion of “ferromagnetic scaling region” contraposed to the “grey zone” (sketch Fig.3 and main text).
- 11 new citations were added to address the points raised by the Reviewers.

Point-by-point response

Reviewer #1

The authors of this paper studied an interesting question concerning the scaling invariance and critical exponents of a phase transition in a case called “avoided critical point”. This question may attract deeper investigations in statistical physics to extend the picture of a phase transition provided by the renormalization group to more complicated and realistic materials.

To their purpose, the authors used experimental data they performed on a monolayer (in fact 1.7 ML) of Fe using SEMPA and MOKE techniques which provided them with a phase diagram in the (T, B) plane where T is the temperature and B the applied magnetic field. The diagram is shown in Fig. 1 (and also in Fig. 2d). They stated that the spin of the layer are perpendicular to the film plane with a ferromagnetic interaction between nearest neighbors and a infinite-range dipolar interaction. They argued that the latter exists in the Fe film (they did not explain why but I think this is from their earlier experimental works). This polar interaction is at the origin of the modulated phase a in region around the would-be critical point T_c if this modulated phase does not exist. They deduced T_c which fits the scaling laws with a large number of experimental data (but not all). They found $T_c = 300$ K and critical exponents $\beta = 0.15 \pm 0.03$ and $\delta = 13 \pm 2$. Within errors these exponents are those of the exactly solved 2D Ising model ($\beta = 0.125$, $\delta = 15$). The data which do not fit the scaling law are scaled with the so-called one-variable scaling relation with g functions as shown in Fig. 2c with the 2D Ising critical exponents . The remaining data which do not collapse even with this law were interpreted as those belonging to the grey zone in Fig. 2d. These points that

violate 2D-Ising scaling, whose number is substantial, are assigned to a phase with modulated magnetisation (grey zone). In this modulated phase, the scaling is $M/B \propto L(T)$ (width of modulated domains) as suggested in the paper PRL (2010) from the authors.

They have also simulated the system using the Monte Carlo method with the Hamiltonian (2). Using a slow heating and a slow cooling in a field, they observed a “critical” value of the field b_c below which the modulated phase sets in. The value of T_c depends on the dipolar strength. The fit with the 2D Ising scaling law is observed for $b > b_c$ where the magnetisation is uniform.

The above description is what I see from what the authors stated in the manuscript. Here are my remarks:

Reviewer #1

1. On the presentation: The manuscript is difficult to read. This is because the authors go back and forth on the scenario at many places which are not easy to follow. It would be preferable that they anticipate the scenario once at the beginning of section II and show the results to justify that. I appreciate on the other hand the section Methods which is well presented.

Authors’ response

We understood this comment of the Reviewer as a suggestions to anticipate the theoretical scenario and Monte-Carlo results before the presentation of the experimental outcomes. Moreover, we enlarged the theoretical section to address the next point. Even if the manuscript has become longer, we think that this change facilitates the readability of the paper and better pinpoints the relevance of our findings. [If the comment was misunderstood, we may also consider to revise this choice and revert to the original sequence: “Experimental results” (before) “Theoretical model” (after)].

Reviewer #1

2. On the physics:

a) The authors adjusted the value of the critical temperature T_c in order to realize the best collapsing of data on the 2D Ising scaling curve. To my opinion, there are two possibilities:

(i) The first is that nothing can warrant that such a T_c exists or corresponds to a physical reality because no experimental data in this paper show a sign of a phase transition.

Authors’ response

Actually, the value of T_c was adjusted to realise the best collapsing only for the data computed with Monte-Carlo simulations. For what concerns the experimental data, T_c was extrapolated from the position of a specific peak in the magnetic susceptibility, as explained in the SI. This difference has been clarified in the present version. Apart from this, we basically embrace the perspective (i), which excludes the existence of a critical temperature for the magnetisation, understood in a conventional sense.

Reviewer #1

(ii) The second possibility is that T_c does really exist though no signature has been given here. The fact that scaling law works using its value means that there may be a kind of mixing of Ising criticality and non critical fluctuations due to modulated structure. The critical point is not “avoided” but masked by another kind of fluctuations. We have seen in the literature that the mixing of various kinds of symmetries is possible (for example in fully frustrated XY spin systems such as the Villain’s domino model or the antiferromagnet triangular lattice, there is a mixing of vortex (KT phase) and the two-fold (Ising) chirality, the transition can be fit with slightly modified Ising critical exponents in spite of the KT nature \Rightarrow there was no need to search for a fictive T_c). In the case studied here, the modulated structure can coexist with an Ising long-range correlation (it can be a kind of staggered magnetisation defined using the period of the modulation width at T_c which is experimentally known, namely $L(T_c)$). So the wording in the title and in the text may be misleading if the second possibility is a reality.

Authors’ response

With this comment the Reviewer touched a crucial aspect of our work and gave us the opportunity to review the literature and revise the presentation of our findings. Indeed, the very same symmetries $O(2)$ and \mathbb{Z}_2 coexist also in the model considered by us: in our case the \mathbb{Z}_2 symmetry is explicit in the Hamiltonian given in Eq. 1, while the $O(2)$ emerges in a perturbative Hamiltonian of the striped ground state. In this sense, the situation is simply reversed with respect to frustrated XY models (where the $O(2)$ symmetry is explicit in the Hamiltonian and the \mathbb{Z}_2 degeneracy is emergent) but a similar phenomenology might be expected. The major difference lies on the fact that XY models are defined on a lattice while the striped ground state of our model defines a superlattice, whose positional order is not warranted at finite T . As a consequence, such observables as the staggered magnetisation are also affected by the type of positional order of the patterned phase: rigorous arguments forbid the occurrence of long-range positional order of the striped pattern at any finite T . In particular, this fact leads to exclude the occurrence of a second-order phase transition (either of the Ising universality class or with slightly different critical exponents) for the magnetisation of any sublattice.

On the other hand, orientational order of stripes is expected to evolve according to a Berezinskii-Kosterlitz-Thouless (BKT) scenario, in line with the emergent O(2)-symmetric effective Hamiltonian (Refs. 17, 18, 21). Though our samples can be considered good candidates to observe such a transition, a similar investigation is beyond the scope of the present study. Due to limitations in the acquisition time of our SEMPA apparatus, an image containing a sufficiently large number of stripe domains to evidence a power-law decay of orientational correlations cannot be recorded. But this, indeed, represents a stimulating project for future investigations.

This comment motivated the following major changes:

- The theoretical part was modified to better clarify the lack of positional order of the striped pattern (in the grey zone).
- The analogies/differences with respect to the frustrated XY models are now mentioned in the section “Conclusions”, citing some related literature (Ref. 59-64).
- Still in the Conclusions, open issues related to the “grey” zone are more thoroughly discussed – including a possible detection of a BKT phase transition from a nematic phase of magnetic domains to a disordered phase.

We thank the Reviewer for raising these interesting points and putting the analogies with frustrated XY models under our attention.

Reviewer #1

b) The Hamiltonian is for an Ising model with spins pointing in the direction perpendicular to the film thickness. This assumption eliminates the second term in the dipolar interaction, retaining only the antiferromagnetic term. My question is: what is the value of the crystalline Fe spin? is there any reason to suppose that the Fe spins are of the Ising type? Is there any anisotropy along the z direction to justify such an assumption?

Authors’ response

The fact that few atomic layers of Fe on Cu(001) are magnetised out of plane is well-established in the literature of ferromagnetic thin films. We added a reference to a ferromagnetic-resonance investigation of the same systems (Ref.43) in which the total out-of-plane anisotropy was estimated to be equivalent to a magnetic field of 4 – 5 kOe. All our previous experimental observations (e.g., Refs.41,42,49) are consistent with a magnetisation pointing perpendicularly to the film plane for Fe thicknesses comparable to the ones used in this work.

As correctly understood by the Reviewer, such a condition allows simplifying the dipole-dipole interaction as follows:

$$\mathcal{H}_{\text{dip}} = \frac{\mu_0}{4\pi} \left[\frac{\vec{\mu}_1 \cdot \vec{\mu}_2}{r_{12}^3} - 3 \frac{(\vec{\mu}_1 \cdot \vec{r}_{12})(\vec{\mu}_2 \cdot \vec{r}_{12})}{r_{12}^5} \right] \simeq \frac{\mu_0}{4\pi} \frac{\mu_1^z \mu_2^z}{r_{12}^3} \quad (1)$$

where $\vec{\mu}_1$ and $\vec{\mu}_2$ are two interacting elementary dipoles and \vec{r}_{12} their relative position. Note that this interaction – of magnetostatic origin – is always present and in this configuration is antiferromagnetic for each pair of interacting dipoles. Therefore, in sufficiently large 2D systems magnetised out of plane the formation of domains is always energetically convenient, for any ratio J/g (see literature below). We recall this in response to the general doubt of the Reviewer concerning the existence of dipolar interaction in this system.

Reviewer #1

c) In the phase space (T, B) where the spin configuration is ferromagnetic with B not zero (the blue horizontal line in Fig. 1), we can’t talk about “critical” region because, strictly speaking, ferromagnets under an applied field does not make a phase transition.

Authors’ response

We renamed as *ferromagnetic scaling region* the portion of the (T, B) in which equations of state of ferromagnets – like Eqs. 2 and 3 – are obeyed, contraposed to the “grey zone”. We hope that the Reviewer will find this wording more appropriate.

Reviewer #1

d) On the simulation: The dipolar term is not a simple spatial “lattice sum” because each term of the sum depends on the instantaneous value of each pair $S_i \cdot S_j$ (+1 or -1) during the simulation. For each spin pair, this value changes with time-evolution by thermal fluctuations. Can the authors explain how the Ewald’s sum is performed taking into account such a real instantaneous spin configuration at each MC sweep? Also, when J/g is very large, the dipolar effect decreases, the spin configuration should tend to ferromagnetic. Thus, the interesting region is that of small J/g .

Authors’ response

We added a classical reference about MC simulations on Ising systems with dipolar interactions, to help the reader

not familiar with the application of the Ewald sums technique to this kind of systems (Ref.66). The use of Ewald sums in lattice systems involves simplifications that are absent for dipolar fluids commonly studied. Indeed, in this particular case the energy variations (needed for MC sampling) can be written in terms of effective interactions which become simple lattice sums, so they are calculated only once at the beginning of each simulation (see section IV, Eq.(23) of the cited reference). Regarding the influence of J/g , it has been proved that the ground state of this system is always modulated, no matter the value of J/g (see Refs.33-36 or A. Giuliani *et al.*, PRB **74**, 064420 (2006)). As J/g increases, the only effect is an increase of the modulation length. Hence, as long as the linear system size is larger than the modulation length, the system cannot be considered ferromagnetic. For the present ultrathin films ($J/g \sim 100$) this is always the case, so the dipolar interactions cannot be neglected for any value of J/g . In particular, small values of this ratio have been already investigated thoroughly in the literature (see, e.g., Refs.19,28,34-38), while the restoration of Ising criticality addressed by us – to the best of our knowledge – was overlooked so far.

Reviewer #1

e) I suggest the authors to add a snapshot of the spin configuration in the modulated region to show the dipolar effect.

Authors' response

Following the Reviewer's suggestion, we added three representative snapshots (Figs.2c) of Monte-Carlo simulations performed for $J/g = 10$ and $b/g = 0.1$ along with *qualitatively* equivalent spin configurations of the experimental system imaged with SEMPA (Figs.2d). The corresponding magnetisation data obtained for $b/g = 0.1$ were also added in Figs.2a,b.

In conclusion, while the subject is interesting I found that a number of points should be clarified before reconsidering the manuscript for publication.

Reviewer #2

The authors study and discuss the (T, B) phase diagram of Fe films. This phase diagram presents a paramagnetic phase, a ferromagnetic phase, and a modulated domain phase, induced by the dipole-dipole interaction. The authors report critical fluctuations associated to the paramagnetic-ferromagnetic transition over many decades of the scaled variables in spite that the corresponding Curie point is actually avoided, as they explicitly show. They compare their results with numerical predictions of a 2d-Ising model with dipolar interactions.

Although the phenomenon of avoided criticality has been described in previous works, as discussed in the introduction of the manuscript, it has not been considered before for this “model” experimental system. I consider the experimental results are impressive and the analysis is simple and rather convincing. In addition, the quality of the presentation is good: the main message of the paper is simple and clear cut, and of general interest. Understanding the details will surely motivate further research.

I would like to recommend this paper for publication in Nature Communications, after the authors address the following questions/suggestions.

Reviewer #2

1) The critical scalings can also be conveniently written in terms of characteristic lengths, by using the corresponding ν exponent:

$$M(T, B) \sim B^{1/\delta} F_1(B^{-\nu/\beta\delta}/\tau^{-\nu}) \quad (2)$$

or

$$M(T, B) \sim \tau^\beta F_2(B^{-\nu/\beta\delta}/\tau^{-\nu}) \quad (3)$$

I wonder if the authors have tried to check for the following relations (latex formulas):

$$M(T = T_c, B) \sim B^{1/\delta} F_3(B^{-\nu/\beta\delta}/L(T_c)) \quad (4)$$

and

$$M(T, B = 0) \sim \tau^\beta F_4(\tau^{-\nu}/L(T)) \quad (5)$$

when we approach the modulated phase at $T = T_c$ or $B = 0$ respectively, and where $L(T)$ is the period of the modulated phase. (F_1, F_2, F_3, F_4 are just scaling functions. I am omitting the \pm differentiation for simplicity and I

mean $\tau = |T - T_c|/T_c$.

In other words, my question is whether $L(T)$ just acts as a cut-off for the paramagnetic-ferromagnetic divergent correlation length (or susceptibility), inducing the above scalings. Please comment.

Authors' response

We appreciated these suggestions by the Reviewer because they set the viewpoint from the perspective of characteristic length scales, which play a crucial role in spontaneous formation of modulated phases as well as in conventional and avoided critical phenomena. In fact, the most natural way to think of the patterned phase would be as a mosaic of finite-size Ising systems (i.e., domains). Along this line, one would conclude that the only difference with respect to the ideal infinite film is that the size of domains acts in the grey zone as a cut-off for the correlation length and therefore the magnetisation inside the domains behaves similarly to that of a finite lattice. We addressed explicitly this issue at the end of the section “Spatial period of the modulated phase” and devoted to it a new full section in the SI, named “Scaling and characteristic lengths”. There we provide arguments to show that the fulfillment of the scaling $M(T, B) \sim B \cdot L$ in the “grey zone”, observed experimentally, is not compatible with the realization of the scaling suggested by the Reviewer with the scaling functions F_3 and F_4 . Furthermore, we performed additional Monte-Carlo simulations for the 2D unfrustrated Ising model to obtain the finite-size scaling function for the magnetisation. Such calculations clearly show that the behavior of M in the grey zone cannot simply be ascribed to the finite size of magnetic domains (Fig.S7).

The scaling plots defined by F_1 (Fig.S5) and F_2 (Fig.S6) have been added in the supplemental material. In agreement with expectations, collapsing is realized for the coloured data points, falling in the “ferromagnetic scaling region” of the (T, B) plane, while it is not realized for the data points falling in the grey zone.

Reviewer #2

2) Please comment about the disorder in the samples. Does it have any kind of impact on the obtained results?

Authors' response

The main origin of disorder in our samples is due to local fluctuations of the Fe thickness. Such fluctuations are known to produce pinning of domain walls or simply hinder their motion and thus prevent the system from adjusting L to its equilibrium value. A sentence has been added in the discussion of Fig.6g, in relation to the bifurcation of the estimates of L obtained upon heating and cooling.

Reviewer #2

3) Is there a theoretical prediction for the (B, T) phase diagram to compare with the experimental one? If so, does the model of Eq.(2) contain the minimal ingredients to describe it? Please comment.

Authors' response

We added a reference to a recent work in which a phase diagram qualitatively similar to the one observed in Fe/Cu(001) films has been computed within the mean-field approximation starting from our model Hamiltonian (Ref.38).

Reviewer #2

4) There is a “long-rage” typo in the text.

Authors' response

We thank the Reviewer for detecting this typo that was corrected accordingly.

We hope that the queries raised by the Reviewers have been addressed in a satisfactory way in the present version.

Yours Sincerely,

Alessandro Vindigni (on behalf of the co-authors)

RESPONSE TO THE REVIEWERS

We would like to thank both Reviewers for the constructive criticism and useful suggestions They provided, which obviously result from a careful review.

As a results of the review process, we believe that the manuscript is now improved and conveys the main outcomes of our work more straightly than previously.

A list of the major changes that were made followed by a point-by-point response (blue) to the Reviewers' comments (black) are provided below.

Major changes

- Apart from re-arrangements and different numbering, we changed Figure 1,2,4b in order to comply with the requests of the Reviewers.
- New Monte-Carlo simulations were performed to produce the snapshots in Fig.2c and the corresponding curve in Fig.2a (for $b/g = 0.1$).
- New experimental images (Fig.2d) were added to compare with microscopic Monte-Carlo configurations (snapshots).
- Fig.S4, S5, S6 were added to address the points raised by the Reviewers (including new Monte-Carlo simulations, Figs.S5a,b).
- The section “Theoretical scenario” was enlarged and anticipated with respect to the description of experiments (see the answer to the specific comment of the Reviewer #1).
- We introduced the notion of “ferromagnetic scaling region” contraposed to the “grey zone” (sketch Fig.3 and main text).
- 11 new citations were added to address the points raised by the Reviewers.

Point-by-point response

Reviewer #1

The authors of this paper studied an interesting question concerning the scaling invariance and critical exponents of a phase transition in a case called “avoided critical point”. This question may attract deeper investigations in statistical physics to extend the picture of a phase transition provided by the renormalization group to more complicated and realistic materials.

To their purpose, the authors used experimental data they performed on a monolayer (in fact 1.7 ML) of Fe using SEMPA and MOKE techniques which provided them with a phase diagram in the (T, B) plane where T is the temperature and B the applied magnetic field. The diagram is shown in Fig.1 (and also in Fig.2d). They stated that the spin of the layer are perpendicular to the film plane with a ferromagnetic interaction between nearest neighbors and a infinite-range dipolar interaction. They argued that the latter exists in the Fe film (they did not explain why but I think this is from their earlier experimental works). This polar interaction is at the origin of the modulated phase a in region around the would-be critical point T_c if this modulated phase does not exist. They deduced T_c which fits the scaling laws with a large number of experimental data (but not all). They found $T_c = 300$ K and critical exponents $\beta = 0.15 \pm 0.03$ and $\delta = 13 \pm 2$. Within errors these exponents are those of the exactly solved 2D Ising model ($\beta = 0.125$, $\delta = 15$). The data which do not fit the scaling law are scaled with the so-called one-variable scaling relation with g functions as shown in Fig.2c with the 2D Ising critical exponents . The remaining data which do not collapse even with this law were interpreted as those belonging to the grey zone in Fig.2d. These points that

violate 2D-Ising scaling, whose number is substantial, are assigned to a phase with modulated magnetisation (grey zone). In this modulated phase, the scaling is $M/B \propto L(T)$ (width of modulated domains) as suggested in the paper PRL (2010) from the authors.

They have also simulated the system using the Monte Carlo method with the Hamiltonian (2). Using a slow heating and a slow cooling in a field, they observed a “critical” value of the field b_c below which the modulated phase sets in. The value of T_c depends on the dipolar strength. The fit with the 2D Ising scaling law is observed for $b > b_c$ where the magnetisation is uniform.

The above description is what I see from what the authors stated in the manuscript. Here are my remarks:

Reviewer #1

1. On the presentation: The manuscript is difficult to read. This is because the authors go back and forth on the scenario at many places which are not easy to follow. It would be preferable that they anticipate the scenario once at the beginning of section II and show the results to justify that. I appreciate on the other hand the section Methods which is well presented.

Authors’ response

We understood this comment of the Reviewer as a suggestions to anticipate the theoretical scenario and Monte-Carlo results before the presentation of the experimental outcomes. Moreover, we enlarged the theoretical section to address the next point. Even if the manuscript has become longer, we think that this change facilitates the readability of the paper and better pinpoints the relevance of our findings. [If the comment was misunderstood, we may also consider to revise this choice and revert to the original sequence: “Experimental results” (before) “Theoretical model” (after)].

Reviewer #1

2. On the physics:

a) The authors adjusted the value of the critical temperature T_c in order to realize the best collapsing of data on the 2D Ising scaling curve. To my opinion, there are two possibilities:

(i) The first is that nothing can warrant that such a T_c exists or corresponds to a physical reality because no experimental data in this paper show a sign of a phase transition.

Authors’ response

Actually, the value of T_c was adjusted to realise the best collapsing only for the data computed with Monte-Carlo simulations. For what concerns the experimental data, T_c was extrapolated from the position of a specific peak in the magnetic susceptibility, as explained in the SI. This difference has been clarified in the present version. Apart from this, we basically embrace the perspective (i), which excludes the existence of a critical temperature for the magnetisation, understood in a conventional sense.

Reviewer #1

(ii) The second possibility is that T_c does really exist though no signature has been given here. The fact that scaling law works using its value means that there may be a kind of mixing of Ising criticality and non critical fluctuations due to modulated structure. The critical point is not “avoided” but masked by another kind of fluctuations. We have seen in the literature that the mixing of various kinds of symmetries is possible (for example in fully frustrated XY spin systems such as the Villain’s domino model or the antiferromagnet triangular lattice, there is a mixing of vortex (KT phase) and the two-fold (Ising) chirality, the transition can be fit with slightly modified Ising critical exponents in spite of the KT nature \Rightarrow there was no need to search for a fictive T_c). In the case studied here, the modulated structure can coexist with an Ising long-range correlation (it can be a kind of staggered magnetisation defined using the period of the modulation width at T_c which is experimentally known, namely $L(T_c)$). So the wording in the title and in the text may be misleading if the second possibility is a reality.

Authors’ response

With this comment the Reviewer touched a crucial aspect of our work and gave us the opportunity to review the literature and revise the presentation of our findings. Indeed, the very same symmetries $O(2)$ and \mathbb{Z}_2 coexist also in the model considered by us: in our case the \mathbb{Z}_2 symmetry is explicit in the Hamiltonian given in Eq. 1, while the $O(2)$ emerges in a perturbative Hamiltonian of the striped ground state. In this sense, the situation is simply reversed with respect to frustrated XY models (where the $O(2)$ symmetry is explicit in the Hamiltonian and the \mathbb{Z}_2 degeneracy is emergent) but a similar phenomenology might be expected. The major difference lies on the fact that XY models are defined on a lattice while the striped ground state of our model defines a superlattice, whose positional order is not warranted at finite T . As a consequence, such observables as the staggered magnetisation are also affected by the type of positional order of the patterned phase: rigorous arguments forbid the occurrence of long-range positional order of the striped pattern at any finite T . In particular, this fact leads to exclude the occurrence of a second-order phase transition (either of the Ising universality class or with slightly different critical exponents) for the magnetisation of any sublattice.

On the other hand, orientational order of stripes is expected to evolve according to a Berezinskii-Kosterlitz-Thouless (BKT) scenario, in line with the emergent O(2)-symmetric effective Hamiltonian (Refs. 17, 18, 21). Though our samples can be considered good candidates to observe such a transition, a similar investigation is beyond the scope of the present study. Due to limitations in the acquisition time of our SEMPA apparatus, an image containing a sufficiently large number of stripe domains to evidence a power-law decay of orientational correlations cannot be recorded. But this, indeed, represents a stimulating project for future investigations.

This comment motivated the following major changes:

- The theoretical part was modified to better clarify the lack of positional order of the striped pattern (in the grey zone).
- The analogies/differences with respect to the frustrated XY models are now mentioned in the section “Conclusions”, citing some related literature (Ref. 59-64).
- Still in the Conclusions, open issues related to the “grey” zone are more thoroughly discussed – including a possible detection of a BKT phase transition from a nematic phase of magnetic domains to a disordered phase.

We thank the Reviewer for raising these interesting points and putting the analogies with frustrated XY models under our attention.

Reviewer #1

b) The Hamiltonian is for an Ising model with spins pointing in the direction perpendicular to the film thickness. This assumption eliminates the second term in the dipolar interaction, retaining only the antiferromagnetic term. My question is: what is the value of the crystalline Fe spin? is there any reason to suppose that the Fe spins are of the Ising type? Is there any anisotropy along the z direction to justify such an assumption?

Authors’ response

The fact that few atomic layers of Fe on Cu(001) are magnetised out of plane is well-established in the literature of ferromagnetic thin films. We added a reference to a ferromagnetic-resonance investigation of the same systems (Ref.43) in which the total out-of-plane anisotropy was estimated to be equivalent to a magnetic field of 4 – 5 kOe. All our previous experimental observations (e.g., Refs.41,42,49) are consistent with a magnetisation pointing perpendicularly to the film plane for Fe thicknesses comparable to the ones used in this work.

As correctly understood by the Reviewer, such a condition allows simplifying the dipole-dipole interaction as follows:

$$\mathcal{H}_{\text{dip}} = \frac{\mu_0}{4\pi} \left[\frac{\vec{\mu}_1 \cdot \vec{\mu}_2}{r_{12}^3} - 3 \frac{(\vec{\mu}_1 \cdot \vec{r}_{12})(\vec{\mu}_2 \cdot \vec{r}_{12})}{r_{12}^5} \right] \simeq \frac{\mu_0}{4\pi} \frac{\mu_1^z \mu_2^z}{r_{12}^3} \quad (1)$$

where $\vec{\mu}_1$ and $\vec{\mu}_2$ are two interacting elementary dipoles and \vec{r}_{12} their relative position. Note that this interaction – of magnetostatic origin – is always present and in this configuration is antiferromagnetic for each pair of interacting dipoles. Therefore, in sufficiently large 2D systems magnetised out of plane the formation of domains is always energetically convenient, for any ratio J/g (see literature below). We recall this in response to the general doubt of the Reviewer concerning the existence of dipolar interaction in this system.

Reviewer #1

c) In the phase space (T, B) where the spin configuration is ferromagnetic with B not zero (the blue horizontal line in Fig.1), we can’t talk about “critical” region because, strictly speaking, ferromagnets under an applied field does not make a phase transition.

Authors’ response

We renamed as *ferromagnetic scaling region* the portion of the (T, B) in which equations of state of ferromagnets – like Eqs. 2 and 3 – are obeyed, contraposed to the “grey zone”. We hope that the Reviewer will find this wording more appropriate.

Reviewer #1

d) On the simulation: The dipolar term is not a simple spatial “lattice sum” because each term of the sum depends on the instantaneous value of each pair $S_i \cdot S_j$ (+1 or -1) during the simulation. For each spin pair, this value changes with time-evolution by thermal fluctuations. Can the authors explain how the Ewald’s sum is performed taking into account such a real instantaneous spin configuration at each MC sweep? Also, when J/g is very large, the dipolar effect decreases, the spin configuration should tend to ferromagnetic. Thus, the interesting region is that of small J/g .

Authors’ response

We added a classical reference about MC simulations on Ising systems with dipolar interactions, to help the reader

not familiar with the application of the Ewald sums technique to this kind of systems (Ref.66). The use of Ewald sums in lattice systems involves simplifications that are absent for dipolar fluids commonly studied. Indeed, in this particular case the energy variations (needed for MC sampling) can be written in terms of effective interactions which become simple lattice sums, so they are calculated only once at the beginning of each simulation (see section IV, Eq.(23) of the cited reference). Regarding the influence of J/g , it has been proved that the ground state of this system is always modulated, no matter the value of J/g (see Refs.33-36 or A. Giuliani *et al.*, PRB **74**, 064420 (2006)). As J/g increases, the only effect is an increase of the modulation length. Hence, as long as the linear system size is larger than the modulation length, the system cannot be considered ferromagnetic. For the present ultrathin films ($J/g \sim 100$) this is always the case, so the dipolar interactions cannot be neglected for any value of J/g . In particular, small values of this ratio have been already investigated thoroughly in the literature (see, e.g., Refs.19,28,34-38), while the restoration of Ising criticality addressed by us – to the best of our knowledge – was overlooked so far.

Reviewer #1

e) I suggest the authors to add a snapshot of the spin configuration in the modulated region to show the dipolar effect.

Authors' response

Following the Reviewer's suggestion, we added three representative snapshots (Figs.2c) of Monte-Carlo simulations performed for $J/g = 10$ and $b/g = 0.1$ along with *qualitatively* equivalent spin configurations of the experimental system imaged with SEMPA (Figs.2d). The corresponding magnetisation data obtained for $b/g = 0.1$ were also added in Figs.2a,b.

In conclusion, while the subject is interesting I found that a number of points should be clarified before reconsidering the manuscript for publication.

Reviewer #2

The authors study and discuss the (T, B) phase diagram of Fe films. This phase diagram presents a paramagnetic phase, a ferromagnetic phase, and a modulated domain phase, induced by the dipole-dipole interaction. The authors report critical fluctuations associated to the paramagnetic-ferromagnetic transition over many decades of the scaled variables in spite that the corresponding Curie point is actually avoided, as they explicitly show. They compare their results with numerical predictions of a 2d-Ising model with dipolar interactions.

Although the phenomenon of avoided criticality has been described in previous works, as discussed in the introduction of the manuscript, it has not been considered before for this “model” experimental system. I consider the experimental results are impressive and the analysis is simple and rather convincing. In addition, the quality of the presentation is good: the main message of the paper is simple and clear cut, and of general interest. Understanding the details will surely motivate further research.

I would like to recommend this paper for publication in Nature Communications, after the authors address the following questions/suggestions.

Reviewer #2

1) The critical scalings can also be conveniently written in terms of characteristic lengths, by using the corresponding ν exponent:

$$M(T, B) \sim B^{1/\delta} F_1(B^{-\nu/\beta\delta}/\tau^{-\nu}) \quad (2)$$

or

$$M(T, B) \sim \tau^\beta F_2(B^{-\nu/\beta\delta}/\tau^{-\nu}) \quad (3)$$

I wonder if the authors have tried to check for the following relations (latex formulas):

$$M(T = T_c, B) \sim B^{1/\delta} F_3(B^{-\nu/\beta\delta}/L(T_c)) \quad (4)$$

and

$$M(T, B = 0) \sim \tau^\beta F_4(\tau^{-\nu}/L(T)) \quad (5)$$

when we approach the modulated phase at $T = T_c$ or $B = 0$ respectively, and where $L(T)$ is the period of the modulated phase. (F_1, F_2, F_3, F_4 are just scaling functions. I am omitting the \pm differentiation for simplicity and I

mean $\tau = |T - T_c|/T_c$.

In other words, my question is whether $L(T)$ just acts as a cut-off for the paramagnetic-ferromagnetic divergent correlation length (or susceptibility), inducing the above scalings. Please comment.

Authors' response

We appreciated these suggestions by the Reviewer because they set the viewpoint from the perspective of characteristic length scales, which play a crucial role in spontaneous formation of modulated phases as well as in conventional and avoided critical phenomena. In fact, the most natural way to think of the patterned phase would be as a mosaic of finite-size Ising systems (i.e., domains). Along this line, one would conclude that the only difference with respect to the ideal infinite film is that the size of domains acts in the grey zone as a cut-off for the correlation length and therefore the magnetisation inside the domains behaves similarly to that of a finite lattice. We addressed explicitly this issue at the end of the section “Spatial period of the modulated phase” and devoted to it a new full section in the SI, named “Scaling and characteristic lengths”. There we provide arguments to show that the fulfillment of the scaling $M(T, B) \sim B \cdot L$ in the “grey zone”, observed experimentally, is not compatible with the realization of the scaling suggested by the Reviewer with the scaling functions F_3 and F_4 . Furthermore, we performed additional Monte-Carlo simulations for the 2D unfrustrated Ising model to obtain the finite-size scaling function for the magnetisation. Such calculations clearly show that the behavior of M in the grey zone cannot simply be ascribed to the finite size of magnetic domains (Fig.S5).

The scaling plots defined by F_1 (Fig.S4a) and F_2 (Fig.S4b) have been added in the supplemental material. In agreement with expectations, collapsing is realized for the coloured data points, falling in the “ferromagnetic scaling region” of the (T, B) plane, while it is not realized for the data points falling in the grey zone.

Reviewer #2

2) Please comment about the disorder in the samples. Does it have any kind of impact on the obtained results?

Authors' response

The main origin of disorder in our samples is due to local fluctuations of the Fe thickness. Such fluctuations are known to produce pinning of domain walls or simply hinder their motion and thus prevent the system from adjusting L to its equilibrium value. A sentence has been added in the discussion of Fig.6g, in relation to the bifurcation of the estimates of L obtained upon heating and cooling.

Reviewer #2

3) Is there a theoretical prediction for the (B, T) phase diagram to compare with the experimental one? If so, does the model of Eq.(2) contain the minimal ingredients to describe it? Please comment.

Authors' response

We added a reference to a recent work in which a phase diagram qualitatively similar to the one observed in Fe/Cu(001) films has been computed within the mean-field approximation starting from our model Hamiltonian (Ref.38).

Reviewer #2

4) There is a “long-rage” typo in the text.

Authors' response

We thank the Reviewer for detecting this typo that was corrected accordingly.

We hope that the queries raised by the Reviewers have been addressed in a satisfactory way in the present version.

Yours Sincerely,

Alessandro Vindigni (on behalf of the co-authors)

REVIEWERS' COMMENTS AFTER RESUBMISSION**Reviewer #1**

In the revised version, the authors have taken into account all the remarks raised in my first report. I read the replies of the authors point by point. They agreed with my remarks and have made a great effort to satisfy all of them in the revised manuscript: in particular the new presentation of the paper (theory first, experiments next), the wording at some delicate points, adding a comparison with the frustrated XY model, adding some MC snapshots to compare with the SEMPA image, ...

I think that as far as my remarks are concerned, the revised paper can be published in Nature Communications.

Reviewer #2

I consider the authors have satisfactorily answered the questions, and presented a rather improved version of the manuscript.